# Cryo-EM structures reveal specialization at the myosin VI-actin interface and a mechanism of force sensitivity

**Pinar S Gurel[1,2], Laura Y Kim[2], Paul V Ruijgrok[3], Tosan Omabegho[3], Zev Bryant[3,4], Gregory M Alushin[1]***

[1]Laboratory of Structural Biophysics and Mechanobiology, The Rockefeller University, New York, United States; [2]Cell Biology and Physiology Center, National Heart, Blood, and Lung Institute, National Institutes of Health, Bethesda, United States; [3]Department of Bioengineering, Stanford University, Stanford, United States; [4]Department of Structural Biology, Stanford University, Stanford, United States

**Abstract** Despite extensive scrutiny of the myosin superfamily, the lack of high-resolution structures of actin-bound states has prevented a complete description of its mechanochemical cycle and limited insight into how sequence and structural diversification of the motor domain gives rise to specialized functional properties. Here we present cryo-EM structures of the unique minus-end directed myosin VI motor domain in rigor (4.6 Å) and Mg-ADP (5.5 Å) states bound to F-actin. Comparison to the myosin IIC-F-actin rigor complex reveals an almost complete lack of conservation of residues at the actin-myosin interface despite preservation of the primary sequence regions composing it, suggesting an evolutionary path for motor specialization. Additionally, analysis of the transition from ADP to rigor provides a structural rationale for force sensitivity in this step of the mechanochemical cycle. Finally, we observe reciprocal rearrangements in actin and myosin accompanying the transition between these states, supporting a role for actin structural plasticity during force generation by myosin VI.

DOI: https://doi.org/10.7554/eLife.31125.001

*For correspondence:
galushin@rockefeller.edu

**Competing interests:** The authors declare that no competing interests exist.

## Introduction

Myosin motor proteins are responsible for movement and force generation across multiple scales of biology ranging from muscle contraction to cell migration to intracellular transport (*Cheney and Mooseker, 1992*; *Huxley, 1969*; *Syamaladevi et al., 2012*). Defects in myosin genes have been linked to muscular dystrophies, cardiac disease, cancer, and deafness, highlighting the critical role of myosins in cell function and human health (*Hirokawa and Takemura, 2003*). In efforts to better understand disease mechanisms and develop potential therapeutics, these motors have been the subject of extensive biophysical, biochemical, and structural characterization (*Cope et al., 1996*; *Houdusse and Sweeney, 2016*; *Sweeney and Houdusse, 2010*). However, as the myosin superfamily features over 20 classes, a detailed understanding of each motor and its specific mechanisms remains incomplete (*Sellers, 2000*).

Despite this diversity, the enzymatic mechanism of ATP-dependent force generation on filamentous actin (F-actin) is fundamentally conserved (*Figure 1A*) (*Geeves, 2016*; *Holmes, 1997*; *Lymn and Taylor, 1971*; *Sweeney and Houdusse, 2010*). The motor domain (MD) binds and hydrolyzes ATP, which allosterically produces conformational changes enabling low-affinity engagement with F-actin (*Figure 1A*, Pre-power stroke state). The transient opening of the small switch II loop (swII) adjacent to the nucleotide binding cleft allows phosphate to escape (*Figure 1A*, Pi release

**eLife digest** Like miniature motors, proteins called myosins generate the forces needed for cells to move and for muscles to contract. Myosins use the energy stored in a chemical called ATP to move along filaments made from another protein called actin and produce force. The same part of the myosin protein that binds to and uses ATP also contacts actin. As a myosin protein consumes ATP, it cycles through a series of shape changes to drive the motor protein forward, altering how it interacts with the actin filament in the process.

Although all myosins use ATP in fundamentally the same way, individual members of this protein family have specialized properties that enable them to carry out different roles. It is not clear whether each type of myosin makes unique contacts with the actin filament, which could help determine these properties. Furthermore, mechanical forces can control the activity of myosin motors in ways that are poorly understood.

Gurel et al. have now looked at a family member called myosin VI, which moves in the opposite direction along actin filaments relative to other myosins, to better understand the properties of these proteins. An imaging technique called cryo-electron microscopy (cryo-EM) was used to determine the three-dimensional structure of myosin VI bound to actin at two steps in its cycle. Gurel et al. found that myosin VI formed specific interactions with actin that were very different from another myosin family member called myosin IIc, whose structure bound to actin was already known. In addition, the structural changes observed between the two stages of myosin VI's cycle provided insight into how force could be used to control the motor.

Together these findings give a more detailed picture of how myosins work. They suggest that the surface of myosin that contacts actin can evolve to change the properties of a specific myosin. Studies of other myosins bound to actin will provide further insight into how distinct interactions relate to motor-specific properties. Future studies could also help scientists to understand how mutations in genes for myosins – which have been linked to a number of diseases in humans – alter the way in which myosins interact with actin filaments. This in turn could give insight into how these mutations disrupt the proteins' activities.

DOI: https://doi.org/10.7554/eLife.31125.002

state), triggering the transition to a moderate F-actin binding affinity ADP state (*Figure 1A*, ADP state) accompanied by large-scale rearrangements in the converter region which are propagated through the lever arm to generate the power stroke (*Llinas et al., 2015*). Subsequent ADP release results in the highest affinity actin-myosin interaction (*Figure 1A*, rigor state). Re-binding of ATP into the nucleotide cleft then promotes myosin dissociation from the filament (*Figure 1A*, post-rigor state) and primes the motor for a successive cycle.

Sequence divergence of the MD across the superfamily has modulated the kinetics of the various steps of this cycle to tune biophysical parameters including duty cycle, ATPase rate, and force sensitivity, and enabled regulation by post-translational modifications (*Uyeda et al., 1994*). Significant sequence diversity is found on the surface of the MD which contacts F-actin, suggesting that modulation of this interface may enable optimization of these parameters for different cellular roles (*Berg et al., 2001*). However, until very recently (*von der Ecken et al., 2016, 2015*), the inaccessibility of MD-F-actin complexes to near atomic-resolution structural characterization has been refractory to the detailed mechanistic dissection of this hypothesis.

In this study, we focus on the MD of myosin VI, an unconventional myosin motor unique in its ability to walk 'backwards' towards actin filament pointed ends (*Wells et al., 1999*). The large insert two in the myosin VI lever arm confers reverse directionality to the motor (*Bryant et al., 2007*; *Park et al., 2007*) and the adjacent converter region adopts a unique conformation contributing to the large motor step size (*Ménétrey et al., 2007*; *Ovchinnikov et al., 2011*), while the rest of the MD, including the actin binding domains, retains a high degree of structural similarity to barbed-end directed motors (*Buss et al., 2004*). In contrast to 'rower' myosins (*Leibler and Huse, 1993*) that generate bulk contractile forces along actin filaments through assembly into filaments, myosin VI operates as either a processive dimeric transporter (*Dunn et al., 2010*; *Sweeney et al., 2007*) or as a monomeric tether (*Lister et al., 2004*). Myosin VI functions in endocytosis (*Altman et al., 2007*;

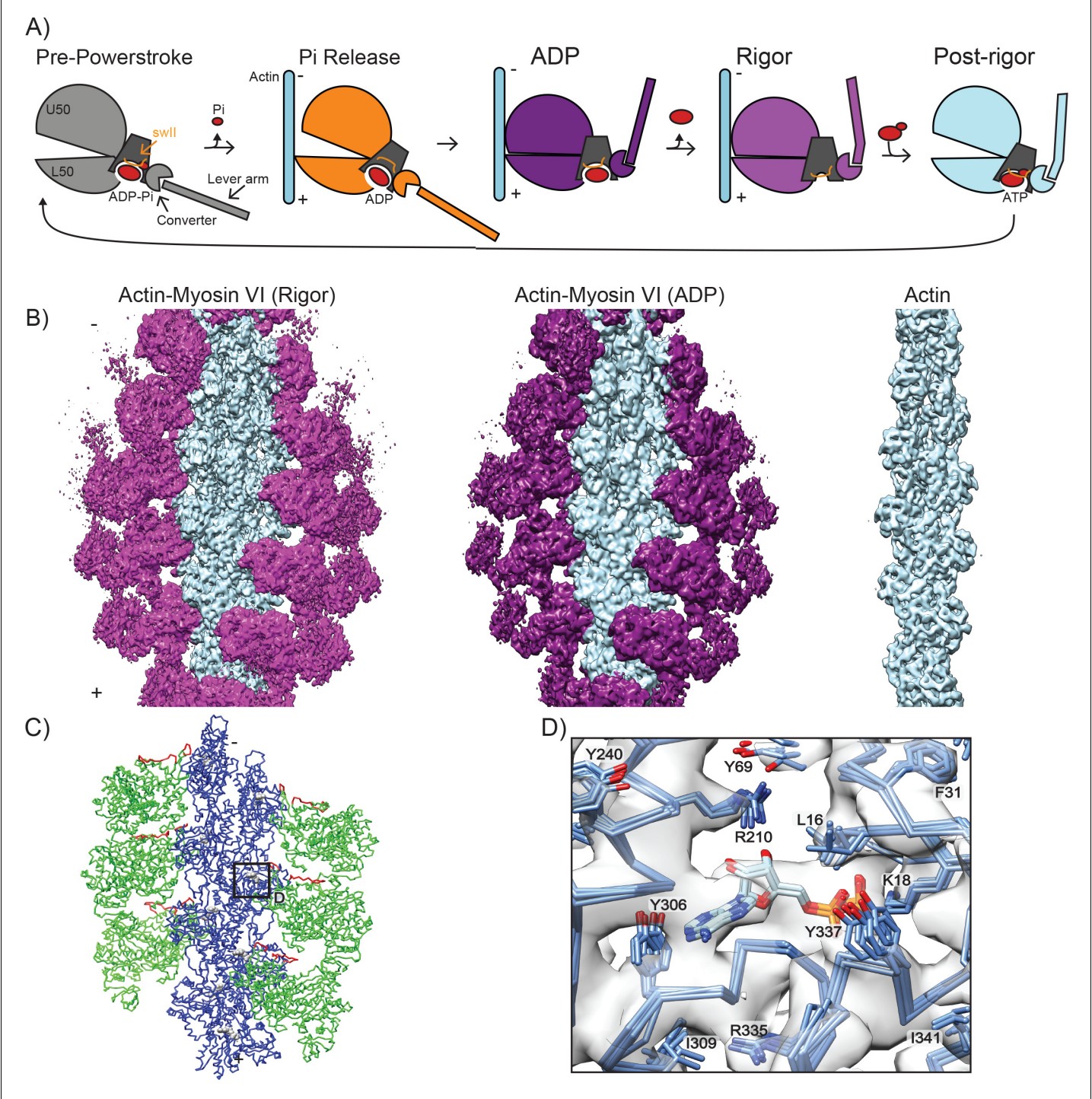

**Figure 1.** High-resolution reconstructions of myosin VI bound to actin. (**A**) Schematic depicting myosin VI states in the force generation cycle. Filament polarity is indicated throughout the paper with pointed end as '-' and barbed end as '+'. Cryo-EM reconstructions of actomyosin VI in the nucleotide-free (rigor) state (left), MgADP (ADP) state (middle), and actin alone (right). Actin, light blue; myosin VI, magenta (rigor) and dark magenta (ADP). (**B**) Atomistic model of actomyosin VI (rigor) colored corresponding to treatment during HR MDFF. Blue (actin), large side chains and backbone atoms subjected to fitting guided by density map; Green (MD), backbone atoms only subjected to fitting guided by density map; Red (MD loop 2 residues 622–636 and HCM loop residues 397–405), density term disabled due to conformational variability; Grey (ADP and Magnesium ion in actin), fixed atoms. (**C**) Superposition of the actin nucleotide-binding cleft from all six actomyosin interfaces in the HR MDFF rigor model, docked into the density map and colored by heteroatom. Large side chains and ADP are displayed.

DOI: https://doi.org/10.7554/eLife.31125.003

*Figure 1 continued on next page*

*Figure 1 continued*

The following figure supplement is available for figure 1:

**Figure supplement 1.** Electron microscopy and resolution assessment.

DOI: https://doi.org/10.7554/eLife.31125.004

*Buss et al., 2001*; *Morris et al., 2002*), intracellular transport (*Inoue et al., 2008*), and stereocilia maintenance (*Avraham et al., 1995*; *Hertzano et al., 2008*; *Melchionda et al., 2001*; *Seiler et al., 2004*), and it has been implicated in deafness (*Ahmed et al., 2003*; *Melchionda et al., 2001*; *Mohiddin et al., 2004*) and cancers (*Dunn et al., 2006*; *Wang et al., 2016*; *Yoshida et al., 2004*). Extensive crystallographic analysis of myosin VI in the absence of actin has produced high-resolution snapshots of many key states (*Llinas et al., 2015*; *Ménétrey et al., 2005*; *Ménétrey et al., 2008*; *Ménétrey et al., 2007*), laying the groundwork for a complete structural description of the mechanochemical cycle of this motor, which serves as a model for the structural biochemistry of the myosin superfamily.

Despite extensive structural and functional characterization, important details of myosin VI mechanism remain unresolved. It remains unclear how precisely phosphate release is coupled to an increase in actin-binding affinity in the ADP state, which is critical for ensuring the MD remains bound to the filament during the power stroke, and how subsequent ADP release further increases actin affinity. The conformation of the ADP state, which dominates the kinetic cycle of the motor and plays a central role in the basic mechanism of force generation (*De La Cruz et al., 2001*; *Robblee et al., 2004*), has not been characterized at high resolution. Additionally, the detailed mechanism by which force asymmetrically influences transitions between the ADP and rigor states is unknown. Mechanically gated acceleration of ADP binding has been reported to enable the motor to switch between anchor and transport functions (*Altman et al., 2004*; *Chuan et al., 2011*; *Robblee et al., 2004*), and force-dependent inhibition of ADP release has also been reported based on single-molecule measurements of monomers (*Oguchi et al., 2008*). Gating of ADP release has been considered as a mechanism for coordinating heads during processive walking (*Dunn et al., 2010*; *Elting et al., 2011*; *Oguchi et al., 2008*), although kinetic measurements of dimers have favored an alternative ATP-gating model (*Sweeney et al., 2007*). Pioneering low-resolution cryo-EM studies suggested that a minor repositioning of the lever arm accompanies the transition from ADP to rigor (*Wells et al., 1999*), but it is unclear how this is coupled to nucleotide-dependent rearrangements within the motor domain, modulation of actin binding affinity, and force sensitivity.

In addition to regulating the mechanochemical cycle, myosin-generated forces have been implicated in influencing actin conformation (*Anson et al., 1995*; *Orlova and Egelman, 1997*; *Prochniewicz et al., 2010*; *Prochniewicz and Thomas, 2001*), the functional implications of which remain unclear. Myosin II filaments induce severing events in F-actin (*Vogel et al., 2013*), and the cryo-EM structure of rigor myosin IIC bound to actin reveals subtle actin conformational changes in response to myosin binding (*von der Ecken et al., 2016*). Moderate-resolution cryo-EM reconstructions of myosin V bound to F-actin in nucleotide-free and ADP states suggest that binding by this motor may induce changes in actin twist, without further modulation of actin conformation during the mechanochemical cycle; however, the level of detail in the maps presented in this study precluded a detailed description of actin conformational changes (*Wulf et al., 2016*). Actin structural rearrangements, such as altered helical twist, were also proposed to play a role in myosin VI motor activity and step size on the basis of early single-molecule tracking and negative-stain electron microscopy studies (*Nishikawa et al., 2002*) that predated our current structural understanding of the myosin VI dimer (*Houdusse and Sweeney, 2016*). It is unknown how myosin VI binding modulates the conformation of F-actin, and if actin assumes multiple conformations throughout the force generation cycle. Furthermore, it remains to be determined if myosin-induced conformational changes in actin are uniform among different myosin classes, or if this is an additional element of motor specialization. High-resolution structural snapshots of myosins in multiple actin-bound states are necessary to clarify this issue.

Here we present cryo-EM reconstructions of myosin VI bound to F-actin in the rigor state at 4.6 Å resolution and the ADP state at 5.5 Å resolution along with corresponding atomistic models. Implementing novel adaptations of the Iterative Helical Real Space Reconstruction (IHRSR) and Molecular

Dynamics Flexible Fitting (MDFF) approaches, we present a detailed model of the myosin VI-F-actin interface, and provide the first structure of myosin VI in the ADP state, to our knowledge the highest-resolution structure of any myosin in this state. We compare our rigor structure to the recent high-resolution structure of the myosin IIC-F-actin interface, finding that while the contact surface is conserved, the specific interactions differ substantially between the two myosins. By comparing our myosin VI-F-actin structures in the ADP and rigor states to each other and pre-existing crystal structures of the motor in actin-free states, we clarify the structural transitions of the force generation cycle and propose a structural mechanism for mechanical regulation of ADP affinity. Finally, by comparing the conformation of actin in the myosin VI-bound ADP and rigor state structures to bare filaments, we find that actin structural deformations accompany motor conformational changes during the force-generation cycle. This suggests that actin structural plasticity plays a role in actomyosin VI activity, an F-actin property which previous studies suggest is also likely to be exploited by other myosins, potentially by distinct mechanisms (*Anson et al., 1995*; *De La Cruz et al., 2001*, *1999*; *Drummond et al., 1990*; *Kim et al., 2002*; *Llinas et al., 2015*; *Noguchi et al., 2012*; *Oztug Durer et al., 2011*; *Prochniewicz et al., 2010*; *Prochniewicz and Thomas, 2001*; *von der Ecken et al., 2016*; *Wulf et al., 2016*).

## Results

### Cryo-EM analysis and atomistic model of the myosin-VI-MD-actin interface

We utilized an engineered myosin VI construct comprising the MD with 1 IQ fused to an RNA-binding L7Ae kink-turn domain (*Figure 1—figure supplement 1A,C*). The L7Ae kink-turn domain is oriented such that RNA-binding extends the lever arm and can tune motor activity (*Omabegho et al., 2017*). Combining datasets with and without RNA bound improved the resolution of our reconstruction considerably, suggesting that RNA binding does not alter motor conformation (*Figure 1—figure supplement 1B*). Thus, we have excluded the engineered regions from our present structural analysis.

For image analysis and 3D reconstruction, we developed a hybrid procedure consisting of initial alignment using an adapted EMAN2/SPARX (*Hohn et al., 2007*; *Tang et al., 2007*) protocol for IHRSR (*Egelman, 2007*), which implements refinement and reconstruction of independent half-datasets to minimize noise bias in resolution estimation and alignment, followed by polishing refinement and reconstruction of the full dataset using FREALIGN (*Lyumkis et al., 2013*). Utilizing this approach, we obtained a 3D reconstruction of the myosin VI MD in the nucleotide-free (rigor) state bound to F-actin at an average resolution of 4.6 Å in the actin filament and bound MD (*Figure 1B* and *Figure 1—figure supplement 1D*). As is often the case with helically symmetric specimens, the level of detail in the map decays radially outward from the center of the filament (*Kucukelbir et al., 2014*). Local resolution analysis suggests a gradient from slightly better than 4 Å in the actin region of the map, where large side-chains are definitively resolved, to around 6 Å resolution in the converter and lever arm, where only the contour of the backbone is visible (*Figure 1—figure supplement 1D,E*). This presents a challenge for analysis, common with cryo-EM reconstructions, where heterogeneity in the map resolution necessitates caution in the generation and interpretation of atomistic models (*Kucukelbir et al., 2014*).

We therefore adapted the molecular dynamics flexible fitting (MDFF) (*Trabuco et al., 2008*) approach to generate a continuous atomistic model which captures high-resolution features in the best-resolved regions of the map by enabling fitting of large side-chains (*Figure 1C*, blue) while avoiding over-interpretation of lower-resolution areas, where the influence of the map was restricted to backbone conformation (*Figure 1C*, green, see Materials and methods for details). To model the MD-actin interface, we assembled eight actin subunits from the cryo-EM structure of the actin-tropomyosin complex (pdb 3J8A [*von der Ecken et al., 2015*]) and 6 MDs from the X-ray structure of nucleotide-free myosin VI (pdb 2BKI [*Ménétrey et al., 2005*]), which were truncated to exclude the converter and lever arm regions. No density was present for two regions of the MD, loop two and the Hypertrophic Cardiomyopathy (HCM) loop, consistent with flexibility, and only the molecular dynamics force field influenced their conformation (*Figure 1C*, red). The resulting atomistic model ('HR', high-resolution) converged well with a molprobity score of 1.44 and a clash score of 0.41

(*Table 1*). Comparison of the HR MDFF rigor model to the crystal structure of myosin VI in rigor-like state in the absence of actin (2BKI) demonstrates increased jaw closure to relieve a clash with the filament (*Figure 1—figure supplement 1F*), highlighting the importance of visualizing the motor bound to actin to determine the structure of the rigor state.

## Interactions at the myosin-actin interface are distinct between different classes of myosins

Cryo-EM structural studies and modelling analyses of diverse actomyosin complexes in strongly-bound states (*Behrmann et al., 2012*; *Fujii and Namba, 2017*; *Lorenz and Holmes, 2010*; *Wells et al., 1999*; *Wulf et al., 2016*) as well as hydroxyl-radical foot-printing studies (*Oztug Durer et al., 2011*) suggest that all myosins studied thus far engage essentially the same surface on F-actin. However, the lack of MD conservation in actin-binding regions suggests differences may exist in how specific interactions with this F-actin surface are formed by different classes of myosins, which could facilitate tuning of motor properties. To assess the level of conservation at the actomyosin interface, we undertook a detailed comparison of the myosin VI rigor HR MDFF model to the recent 3.9 Å structure of the myosin IIC-F-actin rigor complex, as this structure contains side-chain level resolution at the MD-actin interface (*von der Ecken et al., 2016*). For this analysis, we present the

**Table 1.** Data collection and refinement statistics, related to *Figure 1* and *Figure 1—figure supplement 1*

**Data collection**

| EM | Tecnai F20 | | | | |
|---|---|---|---|---|---|
| Voltage (kV) | 200 | | | | |
| Detector | Gatan K2 Summit | | | | |
| Pixel size (Å/pixel) | 1.27 | | | | |
| Electron dose (e⁻/Å²) | 36 | | | | |
| Defocus range (μm) | −1.5 – −3.0 | | | | |
| Conditions | Myosin VI (Rigor) | | Myosin VI (ADP) | | Actin |
| Number of micrographs | 778 | | 377 | | 442 |
| **Reconstruction and Refinement** | | | | | |
| Software | EMAN2/SPARX and FREALIGN | | | | |
| Segments | 56,116 | | 36,114 | | 63,139 |
| Asymmetric Units | 168,348 | | 108,342 | | 189,417 |
| Rise (Å) | 28.06 | | 28.06 | | 28.11 |
| Twist (°) | −166.73 | | −166.69 | | −166.65 |
| Maps | HR Rigor | LPF Rigor | HR ADP | LPF ADP | HR Actin |
| Resolution (Å) | 4.6 | 7.5 | 5.5 | 7.5 | 5.5 |
| Map sharpening B- factor (Å²) | −150 | −150 | −200 | −200 | −350 |
| **Model Building** | | | | | |
| Software | Direx, Coot, MDFF, Phenix | | | | |
| **Validation** | | | | | |
| Molprobity score | 1.44 | 1.78 | 1.40 | 1.42 | 1.63 |
| Clash score | 0.41 | 2.81 | 0.33 | 0.39 | 1 |
| Ramachandran statistics (%) | | | | | |
| Favored | 90.26 | 93.3 | 91.31 | 91.01 | 90.87 |
| Outlier | 1.36 | 1.12 | 1.29 | 1.52 | 0.76 |
| **Structure Deposition** | | | | | |
| PDB Accession Code | 6BNP | 6BNV | 6BNQ | 6BNW | 6BNO |
| EMDB Accession Code | EMD-7116 | | EMD-7117 | | EMD-7115 |

DOI: https://doi.org/10.7554/eLife.31125.005

superposition of all six actomyosin interfaces from the HR MDFF model, facilitating visualization of the clustering of side-chain positions and thereby providing a means of assessing confidence in specific contacts despite the limitations of the map resolution. Particularly well-resolved density regions, such as the actin nucleotide-binding cleft, demonstrate uniform positioning of large side-chains in density peaks, consistent with our resolution assessment (*Figure 1D*). As with myosin IIC, the actomyosin interface is comprised of several myosin surface loops (HCM loop, loop 2, loop 3, loop 4, and helix-loop-helix) located within the upper 50 KD (U50) and lower 50 KD (L50) domains of myosin which interact with subdomains 1 and 3 of one actin, and subdomain 2 of an adjacent actin (*Figure 2A*), supporting overall conservation of the interface architecture (*Behrmann et al., 2012*; *Holmes et al., 2003*; *Rayment et al., 1993*; *Várkuti et al., 2012*; *von der Ecken et al., 2016*).

Regarded as central to all actomyosin interactions (*Kojima et al., 2001*; *Sasaki et al., 2002*), one of the initial contacts between myosin and actin is predicted to occur between the myosin helix-loop-helix (HLH, I525-K550) motif in the L50 domain and an actin hydrophobic patch between actin SD1 and SD3 (*Figure 2B*). In myosin VI we find that the hydrophobic residues P536 and L535 of the HLH are embedded in a groove comprised of I345, L349, and Y143 in actin SD1 (*Figure 2B*), with clear density peaks to support positioning of these side chains. This interface is consistent with hydroxyl radical foot-printing studies demonstrating a hydrophobic interaction between this actin surface and skeletal muscle myosin (*Oztug Durer et al., 2011*). R534 is oriented with its guanidinium group pointing away from the hydrophobic pocket, with the aliphatic portion potentially contributing to the hydrophobic interaction (*Figure 2B*). The HLH for myosin IIC fits into a similar hydrophobic pocket in actin, with a conserved proline (P561) contributing to this interaction. However, the other specific residues involved in the interaction differ substantially (*Figure 2—figure supplement 1B*). In contrast to myosin VI, the myosin IIC HLH is comprised of aromatic side chains, with F560 playing a critical role in the interaction with actin (*von der Ecken et al., 2016*).

Interactions between myosin loop 3 (H551-G576) in the L50 domain and actin SD1 and the D-loop of the adjacent actin subunit form the Milligan contact (*Milligan, 1996*; *Milligan et al., 1990*; *Rayment et al., 1993*), whose precise role in actin engagement is unclear. Studies of other myosins suggest that this interface is formed by complementary charged surfaces rather than specific salt bridges and thus plays only an ancillary role in the generating the high affinity interaction for the rigor state (*Houdusse and Sweeney, 2016*; *von der Ecken et al., 2016*). Indeed, for myosin IIC, this seems to be the case (*Figure 2—figure supplement 1C*). However, the size of loop three varies among myosins, which prior studies have suggested may relate to its prominence in the actomyosin interface (*Van Dijk et al., 1999*). Consistent with this prediction, the large loop 3 of myosin VI likely makes more extensive contacts at this interface than myosin IIC, with MDFF suggesting probable interactions formed between D574 in loop three and K50 in the actin D loop, and E575 and R95 in actin SD2 (*Figure 2C*). The E575 residue is not conserved among any other myosin isoforms, suggesting that this interaction may be specific to myosin VI (*Zhang and Liao, 2012*). For myosin IIC, the actin D loop interaction occurs through E570 in the HLH, whereas for myosin VI the interaction with the actin D loop is likely through D574 in loop 3 (*Figure 2C* and *Figure 2—figure supplement 1C*). An additional unanticipated contact is made by myosin VI R561, which forms a cation-π interaction with Y91 in actin in our model (*Figure 3C*), discussed further in the next section. The actin residues R95 and Y91 have also been implicated myosin strong-binding interactions by hydroxyl radical foot-printing studies (*Oztug Durer et al., 2011*). Some studies suggest the residue homologous to myosin VI S563 also interacts with actin in other myosins (*von der Ecken et al., 2016*; *Zhang and Liao, 2012*), but in our model this residue points away from the interface and could instead potentially play a role in stabilizing loop 3 (not shown).

In addition to the Milligan contact interactions, MDFF suggests myosin VI makes another unique electrostatic interaction with F-actin. E354 in loop 4 (A355-C362) of the U50 domain of myosin likely forms a salt bridge with K328 in actin SD3 (*Figure 2D*) as supported by clear density for side chains in this region. In contrast, D387 in myosin IIC is reported to interact with a similar charged region in actin comprised of K325 and K327 (*Figure 2—figure supplement 1E*). However, our interpretation of the structure suggests that N385 is interacting with the charged actin pocket, since D387 is not oriented in a manner to make contacts with actin in this region (*Figure 2—figure supplement 1D*).

The hypertrophic cardiomyopathy (HCM) loop (T392-P410), which protrudes from the U50 domain, features numerous disease mutations (*Sellers, 2000*), highlighting the importance of this region for stabilizing interactions with actin. In our reconstruction, we observe both an ordered

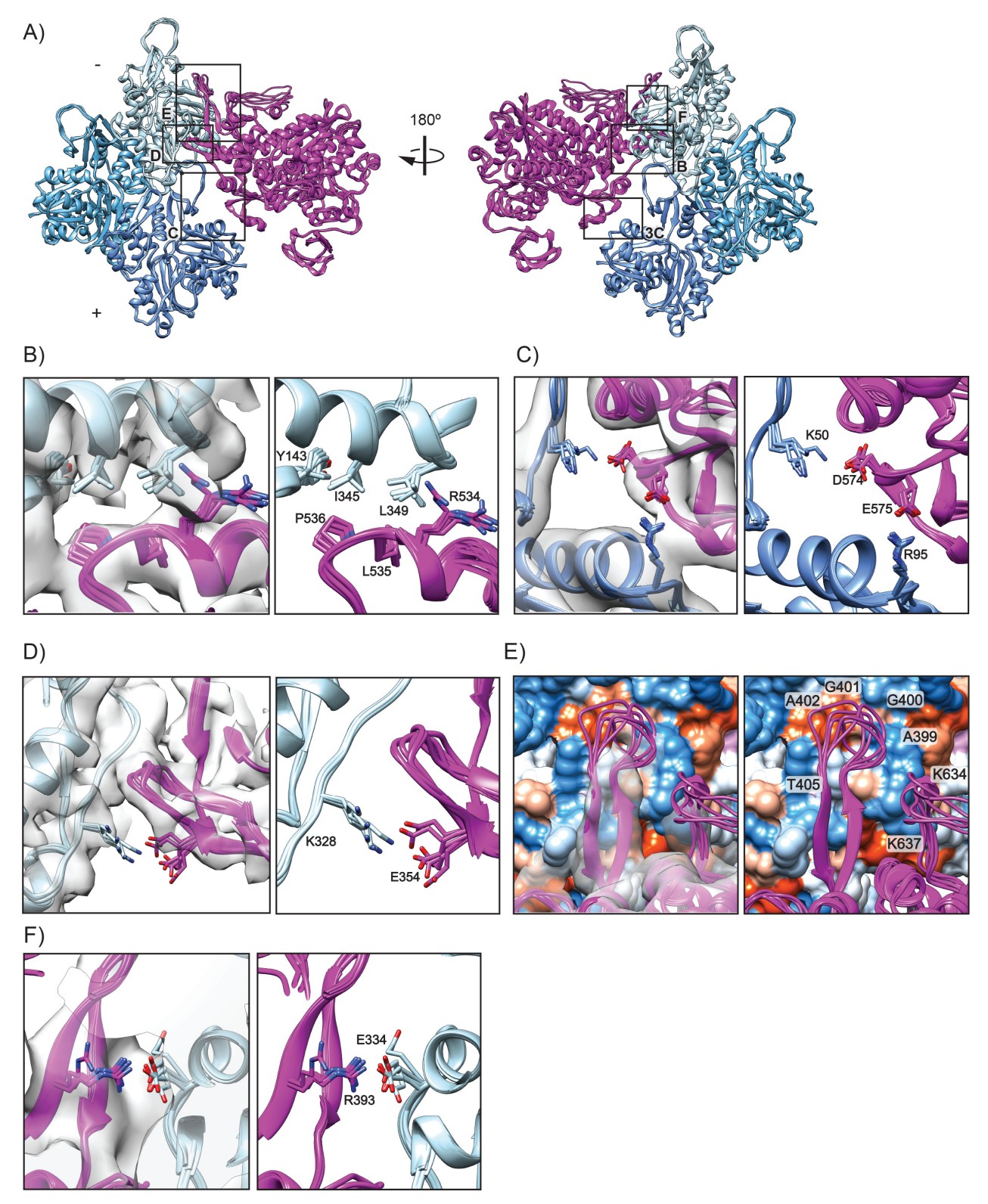

**Figure 2.** Interactions composing the actomyosin VI interface in rigor. (A) All six actomyosin interfaces from the HR MDFF rigor model, superimposed based on the Cα coordinates of the dark-blue actin subunits. MD, magenta; actin subunits, varying shades of blue. (B–D) Detail views of interface contacts suggested by MDFF, colored as in A; EM density map is displayed on left side in transparent grey. (B) Hydrophobic interface between MD HLH and actin SD1/SD3. (C) Milligan contact interactions between MD loop 3 and actin D-loop/SD1. (D) Electrostatic interaction between MD loop 4

*Figure 2 continued on next page*

eLIFE Research article

Biophysics and Structural Biology

*Figure 2 continued*

and actin SD3. (**E**) Interface between MD HCM loop and loop 2 with actin surface, colored by hydrophobicity from most hydrophobic (orange) to most hydrophilic (blue). (**F**) Salt bridge formation between the base of the MD HCM loop with actin SD1.

DOI: https://doi.org/10.7554/eLife.31125.006

The following figure supplement is available for figure 2:

**Figure supplement 1.** Comparison of interactions composing the actomyosin interface between rigor-state myosin IIC and myosin VI.

DOI: https://doi.org/10.7554/eLife.31125.007

segment of the HCM loop, which forms an anti-parallel β-sheet comprising residues 392–396 and 406–410, here referred to as the 'base', as well as a flexible 'tip' for which no density was present in our reconstruction (residues 397–405). As no density was present for the tip, the HR MDFF model exhibits structural variability in this region, and we cannot confidently assign specific orientations to side chains. However, we find that tip residues A399-A402 lie adjacent to a small hydrophobic patch in actin between SD1 and SD2 (*Figure 2E*), which mutagenesis studies in yeast actin have suggested contributes to the strong-binding myosin interface through residue I341 (*Miller et al., 1996*). This is similar to the myosin IIC-actin interface, where the HCM loop docks on to the same hydrophobic patch in actin and is predominantly stabilized by hydrophobic interactions (*Figure 2*- Figure Supplement E). While the myosin IIC HCM loop has weak electrostatic interactions at the tip with R424 fitting into a charged pocket of actin, the myosin VI HCM tip lacks charged residues (*Figure 2—figure supplement 1E*) (*von der Ecken et al., 2016*). A similar electrostatic contact with actin could occur via T405 (*Figure 2E*), a phosphorylation site implicated in regulating directional transport of endocytic clusters (*Buss and Kendrick-Jones, 2008*; *Naccache and Hasson, 2006*).

In contrast with myosin IIC, the ordered myosin VI HCM base likely forms an electrostatic interaction with actin, as MDFF suggests a potential salt bridge between R393 and E334 of actin SD1 (*Figure 2F*). An analogous arginine in myosin IIC, R419, is a disease-related residue important for stabilizing interactions between actin and myosin (*Lorenz and Holmes, 2010*); however, this residue does not interact with F-actin and instead stabilizes the HCM loop through interactions with Y426 on the opposing strand (*Figure 2—figure supplement 1F*)(*von der Ecken et al., 2016*).

Loop 2 (F621-S642) bridges the U50 and L50 domains and has been implicated as the region responsible for initiating binding with actin (*Preller and Holmes, 2013*). While we cannot identify specific interactions due to the structural variability of this loop, for which density was not present, L638-I641 at the base of loop 2 are in close proximity to an actin hydrophobic patch, similar to myosin IIC (*Figure 2E* and *Figure 2—figure supplement 1E*). Neighboring the hydrophobic base, charged loop residues K634 and K637 lie adjacent to an actin acidic patch comprised of D24-D25 and the acidic N-term (*Figure 2E*) which has been reported to be important for weak-binding actomyosin interactions in yeast actin based on mutagenesis analysis (*Miller et al., 1996*). The homologous region in myosin IIC forms an electrostatic belt with this actin acidic patch that stabilizes the base of loop 2, and similar interactions with D24-D25 are predicted for myosin V (*von der Ecken et al., 2016*; *Wulf et al., 2016*). Although the resolution of loop 2 is poor in our map, likely due to flexibility of this segment, myosin VI could potentially form similar types of electrostatic interactions in this region. While higher resolution reconstructions may clarify specific loop two and HCM tip interactions with actin, intrinsic disorder is also likely to limit visualization of these interfaces.

Overall, our analysis reveals a notable lack of conservation at the actomyosin interface between myosin VI and myosin IIC. This is consistent with a model in which the enzymatic core of the MD has been preserved, while a mutable actin-binding surface provides a platform for tuning motor properties. Future structural studies of additional divergent myosin-actin complexes will facilitate the development of a theoretical framework linking specific interface features to biophysical parameters of the MD.

## A unique contact is established upon transition from the ADP state to the rigor state

To investigate the link between myosin nucleotide state, actin binding affinity, and force sensitivity, we obtained a reconstruction of actin bound to myosin VI in the ADP state at an average resolution of 5.5 Å (*Figure 1C*, middle, *Video 1*). The challenge of obtaining high-quality micrographs of this

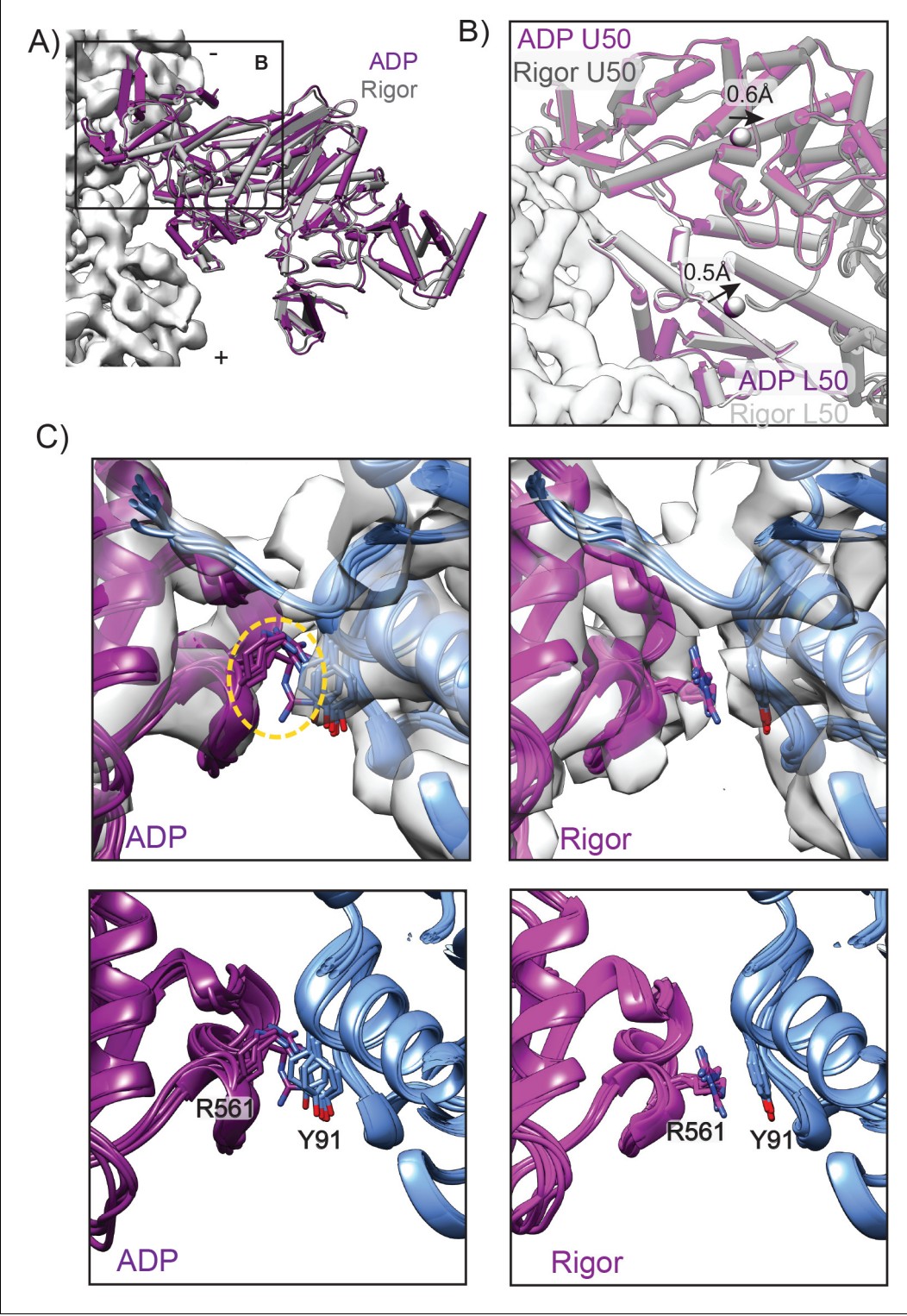

**Figure 3.** A unique contact is established upon transition from ADP to rigor. (**A**) View of the LPF APD MDFF model (dark magenta) and LPF rigor MDFF model superimposed in the reference frame of the actin filament (light gray density). To generate this superposition, the ADP and rigor density maps were aligned, then their corresponding atomistic models were rigid body fit into the aligned maps. (**B**) Minimal actin binding cleft rearrangements are observed between ADP and rigor, superimposed as described in A. ADP U50, magenta; ADP L50, dark magenta; rigor U50, dark grey; rigor L50, light grey; actin density, white. Arrows denote displacements of

*Figure 3 continued on next page*

*Figure 3 continued*

domain centroids (spheres) from ADP to the rigor state. Centroids were determined for U50 (residues 180–206, 229–397, and 405–441) and L50 (residues 467–597 and 638–661) domains. (**C**) MDFF indicates the Milligan contact cation-π interaction between R561 in the MD loop 3 and Y91 in the adjacent actin is absent in ADP (left) but is established upon transition to the rigor state (right), with clear density for these sidechains in the rigor map. For both states, all six actomyosin interfaces in the corresponding HR MDFF model are displayed superimposed on one actin subunit as described in *Figure 2A*. Density maps are displayed in transparent grey in the upper panels. Orange dotted circle indicates absence of density for R561 in the ADP map, while density for Y91 is still present.
DOI: https://doi.org/10.7554/eLife.31125.008

The following figure supplements are available for figure 3:

**Figure supplement 1.** Comparison of ADP and rigor motor nucleotide binding cleft.
DOI: https://doi.org/10.7554/eLife.31125.009
**Figure supplement 2.** Validation of ADP MDFF model.
DOI: https://doi.org/10.7554/eLife.31125.010
**Figure supplement 3.** Interactions composing the actomyosin VI interface in the ADP state.
DOI: https://doi.org/10.7554/eLife.31125.011
**Figure supplement 4.** Comparison of interactions composing the actomyosin VI interface between ADP and rigor.
DOI: https://doi.org/10.7554/eLife.31125.012
**Figure supplement 5.** Transducer rearrangement comparisons between myosins V and VI.
DOI: https://doi.org/10.7554/eLife.31125.013
**Figure supplement 6.** Milligan contact comparisons between myosins IIC, V, and VI.
DOI: https://doi.org/10.7554/eLife.31125.014

lower-affinity actin-bound state limited the number of segments incorporated into this reconstruction. This, along with the ADP state's higher level of flexibility in the converter and lever arm regions suggested by biophysical and modelling studies (*Reifenberger et al., 2009*; *Sun et al., 2007*; *Wulf et al., 2016*) likely limited the overall resolution of this reconstruction.

As with the rigor state reconstruction, the ADP state yielded a multi-resolution map (*Figure 1—figure supplement 1D,E*) with an estimated resolution of 4.7 Å at the actomyosin interface. We observe a clear density peak in the cleft (*Figure 3—figure supplement 1*) that is absent from the rigor density map, as expected for bound ADP; however, the limited resolution precludes detailed modelling of the nucleotide. The ADP-bound Pi Release (PiR) state X-ray structure of myosin VI (4PFO [*Llinas et al., 2015*]) was used as the initial model for MDFF, as it contains ADP in the nucleotide-binding pocket. Because of the overall lower resolution at the interface, only backbone atoms were subject to positioning by the density map during the MDFF simulation, again excluding loop two and the HCM loop. The resulting atomistic model (HR) converged well with a molprobity score of 1.40 and a clash score of 0.33 (*Table 1*). To monitor global rearrangements of the MD between nucleotide states, we used low-pass filtered density maps and MDFF to extend our models for the ADP and rigor state actomyosin complexes to include the converter and lever arm regions (details in Materials and methods). Due to the overall lower resolution of the filtered maps, we limit our analysis to backbone motions and represent these 'LPF' (low-pass filtered) MDFF models as backbone averaged structures instead of a superimposed ensemble (*Video 2*).

As a control for bias imposed by the starting model (4PFO) for the ADP MDFF structures, we also fit the rigor-like myosin VI (PDB 2BKI) structure, which we had previously used as the initial model for our rigor atomistic model, into the ADP state density map. This produced a final

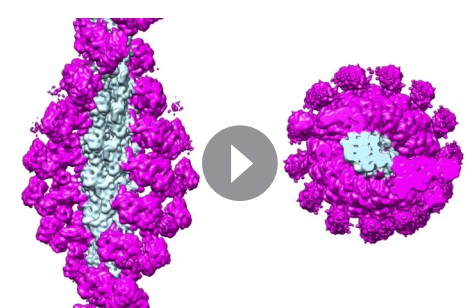

**Video 1.** Morph of cryo-EM reconstructions between ADP and rigor, related to *Figure 1*. Morph from the ADP to the rigor reconstruction, low-pass filtered at 7.5 Å to facilitate visualization of secondary structure rearrangements. To generate this morph, the density maps were aligned to each other. Myosin, magenta; actin, blue.
DOI: https://doi.org/10.7554/eLife.31125.015

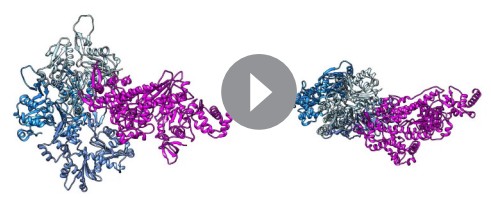

**Video 2.** Morph of atomistic models from ADP to rigor, related to *Figures 2*, *3* and *4*. Morph from the LPF ADP MDFF model to the LPF rigor MDFF model. To generate this morph, the ADP and rigor density maps were aligned, then their corresponding atomistic models were rigid body fit into the aligned maps. Myosin, magenta, three actin subunits, varying shades of blue.
DOI: https://doi.org/10.7554/eLife.31125.016

model (ADP starting from 2BKI) more closely resembling the ADP state starting from 4PFO (Cα RMSD 0.8 Å) than either the 2BKI starting model (Cα RMSD 1.8 Å) or our MDFF model of the rigor state (Cα RMSD 1.1 Å) (*Figure 3—figure supplement 2*, *Table 2*). Regardless of the starting model, MDFF models of the ADP state more closely resemble each other than the rigor state, suggesting our fitting procedure is capturing structural differences between these states that are represented in the maps.

Myosin VI affinity for actin increases as it progresses through the force generation cycle, with the rigor state exhibiting approximately 10-fold higher affinity for actin than the ADP state (*De La Cruz et al., 2001*; *Robblee et al., 2004*). However, it has been unclear how myosin nucleotide state affects actin affinity once the MD has engaged the filament. Prior comparisons of myosin VI crystal structures representing the states preceding (Pre-power stroke, PPS and $P_i$ Release, PiR) and following (rigor-like) the ADP state demonstrated that major actin binding cleft rearrangements, reminiscent of a jaw closing, must occur between PiR and rigor to establish interactions with the actin filament (*Llinas et al., 2015*; *Ménétrey et al., 2005*; *Ménétrey et al., 2008*). While pyrene quenching data (*Llinas et al., 2015*) indicated that cleft closure occurs immediately after $P_i$ release, it remained possible that the ADP state displays an actin-binding cleft structure that is overall closed but distinct from the rigor state, which could be related to the lower affinity of the ADP state compared to the rigor state.

We find very few changes in the actin binding cleft between the ADP and rigor state atomistic models, which are predominantly subtle local rearrangements which do not impact overall cleft closure (*Figure 3A and B*, *Figure 3—figure supplement 3*, and *Figure 3—figure supplement 4*). The Cα RMSD between the U50 and L50 from the rigor and ADP states are 1.5 Å and 1.1 Å, respectively. Additionally, centroid distances of the U50 and L50 between the two states are 0.6 Å and 0.5 Å, demonstrating that there is minimal cleft movement (*Figure 3B*). Atomistic models derived from intermediate-resolution cryo-EM reconstructions for myosin V bound to F-actin also showed minimal cleft rearrangements between the ADP and rigor states (*Wulf et al., 2016*), in agreement with our findings that the major structural changes leading to actin-binding cleft closure must precede the ADP state. As actin binding cleft changes are minimal, alternative mechanisms may also be involved in increasing the affinity of the rigor state. In myosin V, the transducer, a large β-sheet linking the nucleotide-binding cleft to the actin-binding cleft, was reported to adopt a strained conformation in the ADP state, which is relieved upon nucleotide release (*Wulf et al., 2016*). This motivated a model in which an effective increase in actin binding affinity resulted from relief of intramolecular strain in the MD as opposed to a conformational change which modifies contacts with actin. We observe a similar rearrangement of the transducer in myosin VI (*Figure 3—figure supplement 5*); however, we reasoned that subtle modulation of the actin-binding interface could also contribute to differential binding affinity between these states.

Analysis of side chain interactions suggested by MDFF shows that nearly all interactions are likely to be maintained between the two states (*Figure 3—figure supplement 3* and *Figure 3—figure supplement 4*), with the exception of a single residue pair at the Milligan contact. The cation-π interaction between R561 in loop three with Y91 of actin SD2 is not present for the ADP state, suggesting that this contact is likely to form upon the transition from ADP to rigor (*Figure 3C*). Supporting this model, density for R561 is present in the rigor state reconstruction, but notably absent in the ADP state, consistent with R561 being disordered in this state (*Figure 3C*). This cation-π interaction is also absent in the ADP from 2BKI model we generated for validation purposes (*Figure 3—figure supplement 6*). Previous sequence analysis suggests R561 is conserved with only one other human myosin (*Zhang and Liao, 2012*), and a similar interaction is absent in both myosin IIC and myosin V

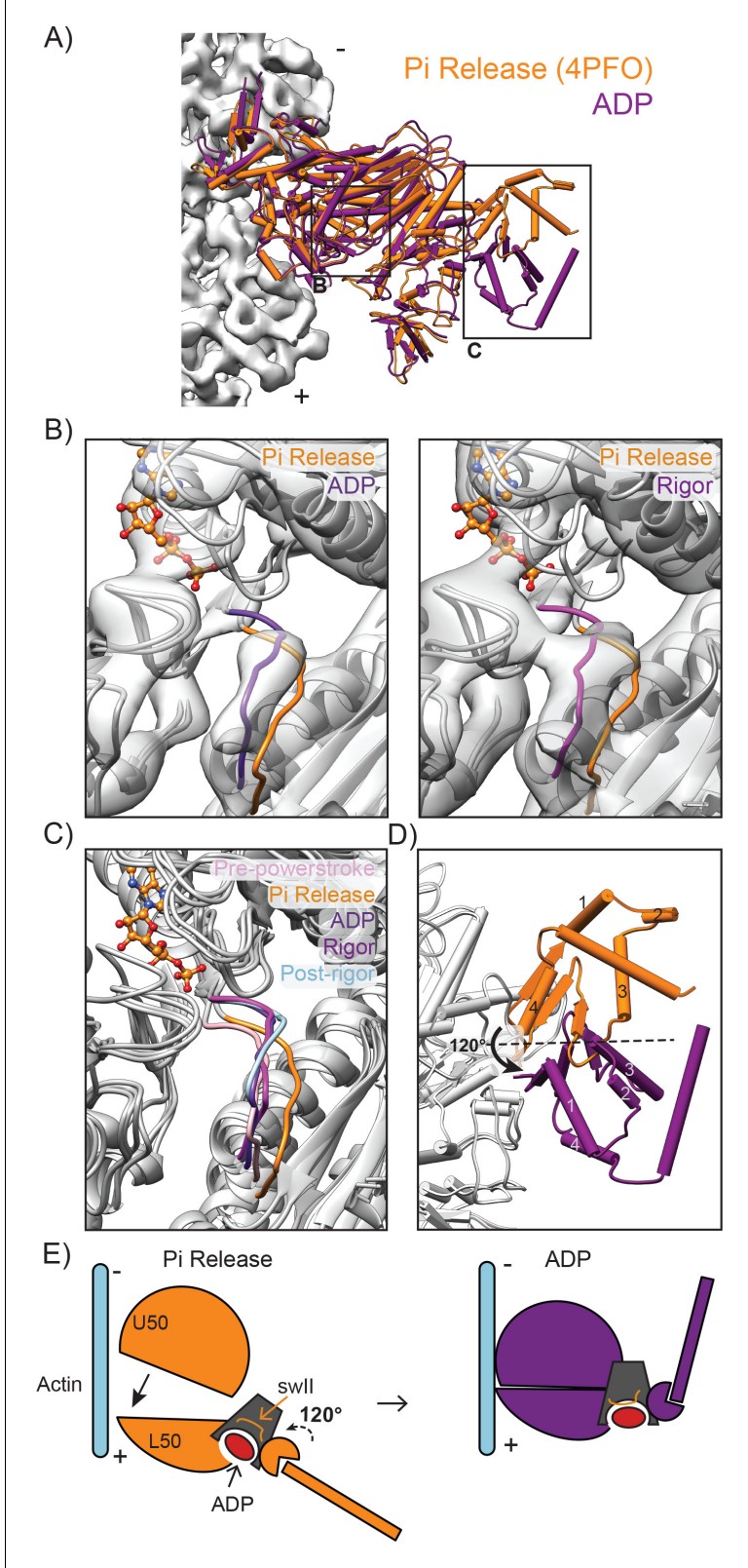

**Figure 4.** The converter adopts the post-power stroke conformation in the ADP state accompanied by switch II closure. (**A**) View of the crystal structure of the PiR state (orange, 4PFO) superimposed on the LPF ADP MDFF model (dark magenta) based on the Cα coordinates of the full motor domain. Actin density is displayed in light grey. (**B**) Comparison of swII orientation between the PiR and ADP (left) and PiR and Rigor (right), superimposed as

*Figure 4 continued on next page*

*Figure 4 continued*
described in A. Density maps for ADP (left) and rigor (right) are displayed. ADP from PiR state is displayed in ball and stick representation and colored by heteroatom. (C) Comparison of swII orientations (various colors) between five states in the force generation cycle. The ADP and rigor maps were aligned, then their corresponding atomistic models were rigid body fit into the aligned maps. MDs from crystal structures were then superimposed based on the Cα coordinates of the full motor domain utilizing ADP as the reference for PiR and PPS, and rigor as the reference for Post-rigor. ADP from PiR is displayed in ball and stick representation and colored by heteroatom. (D) Magnified view of the 120° rotation of the converter and lever arm upon the transition from the PiR state to the ADP state as displayed in A. (E) Schematic depicting the myosin VI transition from PiR to ADP.
DOI: https://doi.org/10.7554/eLife.31125.018
The following figure supplement is available for figure 4:

**Figure supplement 1.** Comparison of converter modeling in Pi Release vs. ADP.
DOI: https://doi.org/10.7554/eLife.31125.019

structures (*Figure 3—figure supplement 6*), suggesting that this interaction may have evolved to support the specialized properties of myosin VI. Formation of this contact could play a role in increasing affinity for actin in the ADP to rigor transition of myosin VI; future high-resolution structural studies will be required to establish if analogous minor adjustments to the filament binding interface play a role in myosin V, as well as other myosins during this transition.

## The converter adopts the post-power stroke conformation in the ADP state accompanied by swII and actin-binding cleft closure

The initiation of force generation occurs once the motor hydrolyzes ATP but has not yet released phosphate, leading to a weak interaction with actin termed the pre-power stroke state (PPS, *Figure 1A*). Crystallographic analysis revealed a subsequent phosphate release state (PiR) representing the state immediately preceding the ADP state where phosphate has been released through a proposed escape tunnel (*Llinas et al., 2015*), but the lever arm has not yet swung (*Figure 1A*). Comparison of the PiR structure to our ADP structure thus facilitates a detailed analysis of the structural transitions accompanying the primary power stroke of myosin VI (*Figure 4A*).

The switch II loop (swII) plays an important role in arranging and stabilizing the myosin nucleotide-binding pocket. The PiR structure revealed that swII adopts an open conformation in this state, opening a path that would allow phosphate escape from the nucleotide binding pocket (*Llinas et al., 2015*). This observation lead to the hypothesis that swII would transiently open only in the PiR state, closing immediately after to prevent phosphate re-binding and thereby enforcing the forward directionality of the mechanochemical cycle (*Llinas et al., 2015*). Consistent with this model, we find that in both the ADP and rigor states, as with the PPS and post-rigor state structures, swII adopts a closed conformation when compared to the PiR state (*Figure 4B and C*).

To execute the power stroke, movement of the nucleotide binding cleft is propagated via the transducer, the relay helix, and the SH1 helix, leading to converter rearrangements which amplify these subtle motions into the swing of the lever arm. Unlike other myosins, the myosin VI PPS converter adopts an unusual conformation and must undergo rearrangements to transition into the rigor state (*Ménétrey et al., 2007*; *Ovchinnikov et al., 2011*). However, it has been unclear whether the ADP converter also adopts this unique PPS conformation and how converter rearrangements are propagated into lever arm movement prior to and after the ADP state. By comparing all previously

**Table 2.** RMSD of Cα positions between models, related to *Figure 2*.

|  | Rigor | ADP | ADP starting from 2BKI | 2BKI (rigor-like) | 4PFO (PiR) |
|---|---|---|---|---|---|
| Rigor |  | 1.3 | 1.1 | 1.6 | 4.4 |
| ADP | 1.3 |  | 0.8 | 1.9 | 3.8 |
| ADP starting from 2BKI | 1.1 | 0.8 |  | 1.8 | 3.9 |
| 2BKI (rigor-like) | 1.6 | 1.9 | 1.8 |  | 4.5 |
| 4PFO (PiR) | 4.4 | 3.8 | 3.8 | 4.5 |  |

DOI: https://doi.org/10.7554/eLife.31125.017

crystalized converter conformations with our density maps (*Figure 4—figure supplement 1*), we confirm that the major converter rearrangement occurs from PiR to ADP, with the converter adopting a post-power stroke, rigor-like conformation in this state (*Figure 4D*). Our structural data are thus consistent with a model in which the major power stroke is accomplished by a converter rearrangement licensed by cleft closure immediately upon phosphate release (*Figure 4E*) (*Llinas et al., 2015*).

## MD rearrangements facilitating ADP release are accompanied by a lever arm bend which could be regulated by force

Previous studies have predicted that forces propagated through the lever arm can allosterically control ADP release by gating conformational transitions in the motor domain required for nucleotide escape (*Altman et al., 2004*; *Oguchi et al., 2008*). Early low-resolution cryo-EM structures of myosin VI were consistent with this hypothesis, demonstrating that a small lever arm swing (~15–20°) accompanies the transition from ADP to rigor, presumably due to nucleotide-dependent rearrangements in the MD (*Wells et al., 1999*). However, the nature of these rearrangements and the mechanism coupling them to lever arm dynamics remain unclear.

We observe a ~ 30° rotation of the converter around an axis nearly parallel to the actin filament upon the transition from ADP to rigor (*Figure 5—figure supplement 1*, *Video 3*). This repositioning is sterically coupled to nucleotide cleft opening by opposing motions in the SH1 helix, which transitions from an extended to compact conformation, and the long relay helix, which exhibits winding at the end proximal to the converter (*Figure 5A and B*, *Figure 5—figure supplement 2*, and *Video 4*). The relay helix contacts the transducer, which coordinates movement of the switch I (residues 193–205) and N-terminal loops (residues 96–106 and 305–312) away from the nucleotide-binding pocket (*Figure 5A*, *Video 2*, and *Video 5*). The SH1 helix is connected to an unnamed loop we here refer to as the cleft loop (residues 670–681), which unexpectedly displays coherent displacement away from the cleft in the opposite direction (*Figure 5A and B*). This remodeling is accompanied by smaller rearrangements in the P-loop (residues 151–156) and insert 1 (residues 278–303), which likely do not play a major role in this step of the mechanochemical cycle (*Figure 5A*, and *Video 5*).

To examine the coupling between these rearrangements and the lever arm, we grafted the X-ray structure of the ordered segment of the myosin VI lever arm (PDB 3GN4 [*Mukherjea et al., 2009*]) on to the distal end of insert two present in our LPF models (*Figure 5C*) (details in experimental procedures). As was observed in the rigor-like crystal structure (*Ménétrey et al., 2005*), the rigor lever arm displays a prominent bend between insert 2 residues 784 and 785, which our map and model reveals to be absent in the ADP state (*Figure 5C and D*, *Figure 4—figure supplement 1*). Based on these grafted models, bending produces a 30° reorientation of the lever arm, which protrudes off the filament axis in the ADP state but is almost perfectly parallel to the filament in the rigor state, providing a new explanation for early EM observations (*Wells et al., 1999*). The converter rotation and lever arm bending results in a ~ 35 Å displacement of the tip of our modeled lever arm, with a ~ 10 Å projected displacement along the filament axis towards the pointed end (*Figure 5D*). Although a similar magnitude displacement (9 Å) was observed to accompany this sub-step in single-molecule optical trapping assays of full-length monomeric myosin VI (*Lister et al., 2004*), the construct employed in this study featured additional sequence contributing to the lever arm and thus cannot be directly compared to our truncated model.

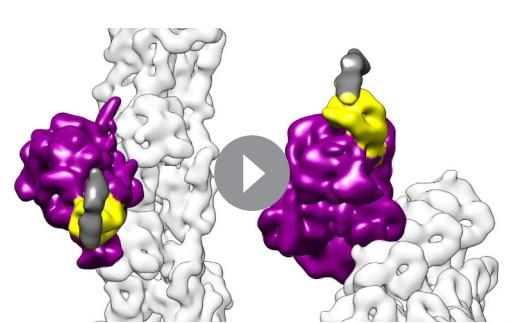

**Video 3.** Morph of converter and lever arm domains between cryo-EM reconstructions of ADP and rigor, related to *Figure 4*. Morph of segmented density maps from the ADP to the rigor reconstruction, low-pass filtered at 7.5 Å aligned in the reference frame of the actin filament. To generate this morph, the full ADP and rigor density maps were aligned, then the segmented maps were aligned to their corresponding full density maps. Motor domain, dark magenta, converter, yellow, lever arm, dark grey. Actin reconstruction is shown in light-grey for orienting the view.

DOI: https://doi.org/10.7554/eLife.31125.024

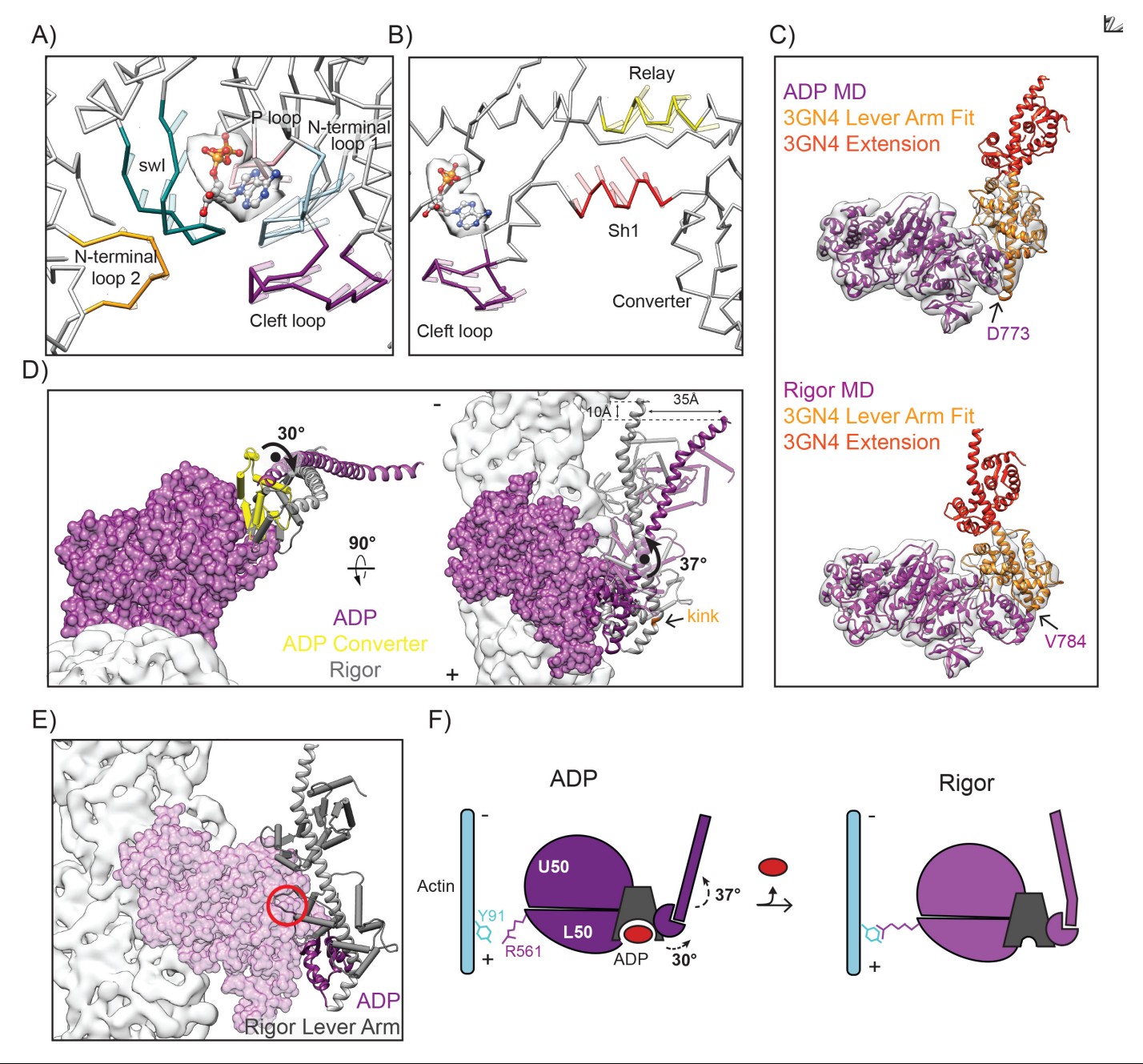

**Figure 5.** Nucleotide release promotes a converter rotation coupled to lever arm bending. (A) Opening of the MD nucleotide-binding cleft depicted by vector traces of Cα displacement from the ADP to the rigor LPF MDFF model of highlighted loops, after aligning the models as described in *Figure 3A*. Displacement vectors are scaled by 1.5 and depicted as transparent rods extending from the ADP LPF MDFF model protein backbone. Coloring is as follows: N-term loop 1 (residues 96–106), blue; N-term loop 2 (residues 305–312), yellow; switch I (residues 193–205), teal; P loop (residues 151–156), pink; cleft loop (residues 670–684), dark magenta, non-highlighted areas, grey. An ADP molecule (ball and stick representation colored by heteroatom) is provided as a visual guide to orient the view. Segmented density from the ADP reconstruction attributable to nucleotide is displayed in transparent grey. (B) Shifted view from A, highlighting the winding of the relay helix (yellow) and translocation of the SH1 helix (red) coupled to opening of the cleft loop (dark magenta). (C) Fit of models with lever arms grafted from crystal structure 3GN4 (orange and red) into their respective density maps filtered at 7.5 Å: ADP (left, magenta), rigor (right, dark magenta). Sites where models were joined are indicated. Orange portion of 3GN4 indicates region that was rigid-body fit into the density maps. Positioning of the red portion is extrapolated from the crystal structure. (D) Converter rotation parallel to filament axis (left) and lever arm bend perpendicular to filament axis (right) between ADP (purple) and rigor (grey). The ADP MD is displayed in surface representation; actin density is light grey. To highlight converter rearrangements, converters are depicted in pipe and plank representation, with the ADP converter in yellow. Extended lever arm models are shown in ribbon representation, with calmodulins depicted in

*Figure 5 continued on next page*

*Figure 5 continued*
transparent pipe and plank. To generate the displayed superposition, the maps of the ADP and rigor state were aligned, then the LPF MDFF models were rigid-body fit into the corresponding map. The extended lever arm model from each state was then superimposed on its corresponding LPF MDFF model based on common Cα coordinates. (**E**) A clash (red circle) between calmodulin (gray pipes and planks) and the MD (transparent purple surface) in the bent state lever arm (gray ribbon) would prevent it from adopting this conformation in the ADP state due to the orientation of the converter (purple ribbon). This analysis was conducted by superimposing the Cα coordinates of the converters (residues 706–773) of the extended-lever arm models shown in C. The converter and MD of the ADP model are displayed. (**F**) Schematic depicting myosin VI transition from ADP to rigor.
DOI: https://doi.org/10.7554/eLife.31125.020

The following figure supplements are available for figure 5:

**Figure supplement 1.** Converter modeling for ADP and rigor states.
DOI: https://doi.org/10.7554/eLife.31125.021
**Figure supplement 2.** EM density map comparison of ADP and rigor SH1 and relay helices.
DOI: https://doi.org/10.7554/eLife.31125.022
**Figure supplement 3.** Calmodulin interactions with myosin VI.
DOI: https://doi.org/10.7554/eLife.31125.023

While our maps and models do not contain atomistic detail in this region, it is tempting to speculate that bending is driven by an electrostatic interaction between negatively charged residue E14 in the proximal light chain bound to insert two and positive residues K736 and R732, which can be seen in the high-resolution rigor-like crystal structure (*Figure 5—figure supplement 3*). A bent lever arm and this interaction are sterically incompatible with the ADP converter position, which would cause severe clashes between the proximal light chain and the MD (*Figure 5E*).

Our models suggest that nucleotide release is coupled to a converter rotation that licenses a lever arm bend in myosin VI, contributing to the displacement observed in previous structural and functional studies of the ADP to rigor transition (*Lister et al., 2004*; *Wells et al., 1999*) (*Figure 5F*). This mechanism is clearly not responsible for the small lever arm swing recently reported to be coupled to ADP release in myosin V, which lacks insert 2 (*Wulf et al., 2016*); rather, it provides a distinct, additional mechanism for myosin VI to reposition the lever arm between the ADP and rigor states. Myosin VI thus seems to have evolved unique conformational changes contributing to both the major power stroke, in which a rearrangement of the converter leads to a larger stroke size than would otherwise be obtained (*Ménétrey et al., 2007*), and the subsequent ADP release sub-step, which amplifies converter rearrangements along the filament axis with a straight-to-bent transition in the lever arm. Furthermore, we propose that force could gate nucleotide engagement by regulating lever arm bending and the associated converter repositioning, with differential effects depending on the geometry (see Discussion).

## Actin rearrangements accompany the myosin VI force generation cycle

F-actin has the capacity for structural polymorphism (*Galkin et al., 2010*), and has been observed to adopt distinct conformational states when in complex with several binding partners, notably becoming severely distorted when decorated with the severing factor cofilin (*McGough et al., 1997*). Extensive biochemical and biophysical studies have suggested that myosin binding induces actin structural rearrangements and that actin structural plasticity is

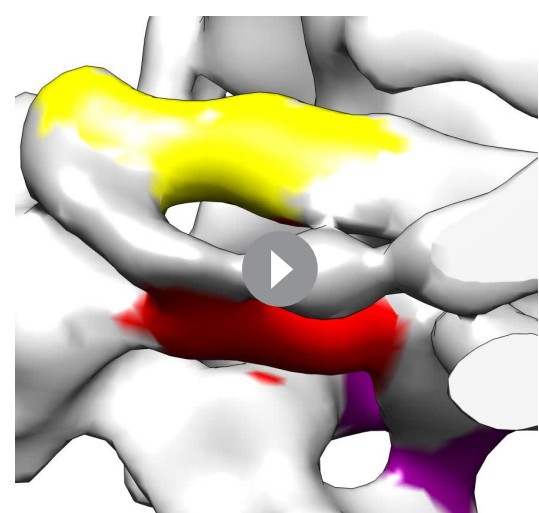

**Video 4.** Morph of the relay and SH1 helices between cryo-EM reconstructions of ADP and rigor, related to *Figure 4*. Morph from the ADP to rigor reconstruction, low-pass filtered at 7.5 Å, focusing on the relay helix (yellow) and SH1 helix (red). To generate this morph, density maps were aligned to each other.
DOI: https://doi.org/10.7554/eLife.31125.025

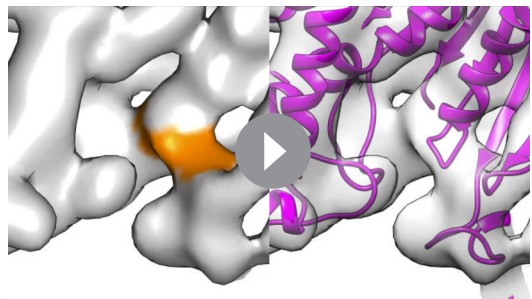

**Video 5.** Morph of the nucleotide-binding cleft between cryo-EM reconstructions of ADP and rigor, related to *Figures 3* and *4*. Morph from the ADP to rigor reconstruction, low-pass filtered at 7.5 Å focusing on the myosin VI nucleotide binding cleft with density corresponding to ADP nucleotide colored orange (left panel). To generate this morph, density maps aligned to each other. Right panel includes the LPF rigor MDFF model rigid body fit into the rigor density map.
DOI: https://doi.org/10.7554/eLife.31125.026

critical for proper myosin activity (*Anson et al., 1995*; *Drummond et al., 1990*; *Kim et al., 2002*; *Nishikawa et al., 2002*; *Noguchi et al., 2012*; *Oztug Durer et al., 2011*; *Prochniewicz et al., 2010*; *Prochniewicz and Thomas, 2001*). Several recent structural studies have described subtle conformational changes in actin when bound by nucleotide-free myosin motor domain (*von der Ecken et al., 2016*; *Wulf et al., 2016*). Differences between conformations of actin filaments decorated with myosin V in the nucleotide-free and ADP states have also been described at intermediate resolution (*Wulf et al., 2016*). Furthermore, an indirect reporter of actin conformation based on changes in pyrene fluorescence quenching (*De La Cruz et al., 2001*; *De La Cruz et al., 1999*; *Kim et al., 2002*; *Llinas et al., 2015*; *Prochniewicz et al., 2010*; *Wulf et al., 2016*) has suggested that actin rearrangements accompany transitions between different states in the myosin mechano-chemical cycle for myosin V and VI. However, the absence of high-resolution structures of the same myosin in multiple states bound to actin has hampered direct visualization of these rearrangements and an interpretation of their functional relevance during force generation.

As we expected actin conformational changes to be subtle, we obtained a reconstruction of F-actin alone at 5.5 Å (*Figure 1B*, right), as well as a corresponding MDFF model to control for error in micrograph pixel size calibration and differences in processing procedures with previously reported structures (*Galkin et al., 2015*; *von der Ecken et al., 2015*). The lower resolution of this reconstruction vs. those bound to myosin suggests that myosin binding may rigidify the filament and reduce inherent conformational flexibility (*Galkin et al., 2012*). The myosin-bound density maps were then aligned in the reference frame of the actin-alone reconstruction, followed by re-docking of the corresponding MDFF models, a procedure which we found revealed regular patterns of actin protomer rearrangements which were masked when the MDFF models were superimposed based on the Cα coordinates of individual actin subunits (data not shown).

The refined helical parameters are essentially identical for all three filament states we report here (*Table 1*), in contrast to both myosin IIC and myosin V, where myosin binding has been reported to induce a 0.5–0.8° change in azimuthal rotation (*von der Ecken et al., 2016*; *Wulf et al., 2016*). Despite this preservation of filament architecture, the actin protomer adopts a unique conformation in each of the three states, with local deformations occurring at the actomyosin interface mediated by the D loop and, intriguingly, at distal lateral contacts in the interior of the filament mediating the interaction between the two strands of the actin filament (*Figure 6*, *Figure 6—figure supplement 1*, *Videos 6* and *7*).

The D-loop is a flexible region of actin SD2 that can adopt a range of conformations to facilitate longitudinal contacts during actin polymerization (*Dominguez and Holmes, 2011*; *Galkin et al., 2010*; *Otterbein et al., 2001*) and engage with actin binding partners (*Dominguez and Holmes, 2011*). As discussed earlier, the D-loop forms interactions with myosin VI loop three as a part of the Milligan contact. Consistent with what has previously been reported for myosin IIC (*von der Ecken et al., 2016*) and IE (*Behrmann et al., 2012*), this flexible loop also shifts upon myosin VI binding, orienting slightly towards the MD (*Figure 6B–C* and *Video 3*). Both the ADP and rigor states exhibit excursions of the D-loop relative to the unbound state, with a Cα RMSD of 1.0 Å in both cases (*Figure 6D*). The Cα RMSD between the ADP and rigor states is also of a similar magnitude (0.9 Å, *Figure 6D*), demonstrating that the D-loop adopts distinct conformational states as the myosin force generation cycle proceeds.

The historically named hydrophobic plug (H-plug, residues 263–273) adjoins three actin subunits within the filament lattice, fitting into the groove created by the interface between two actin

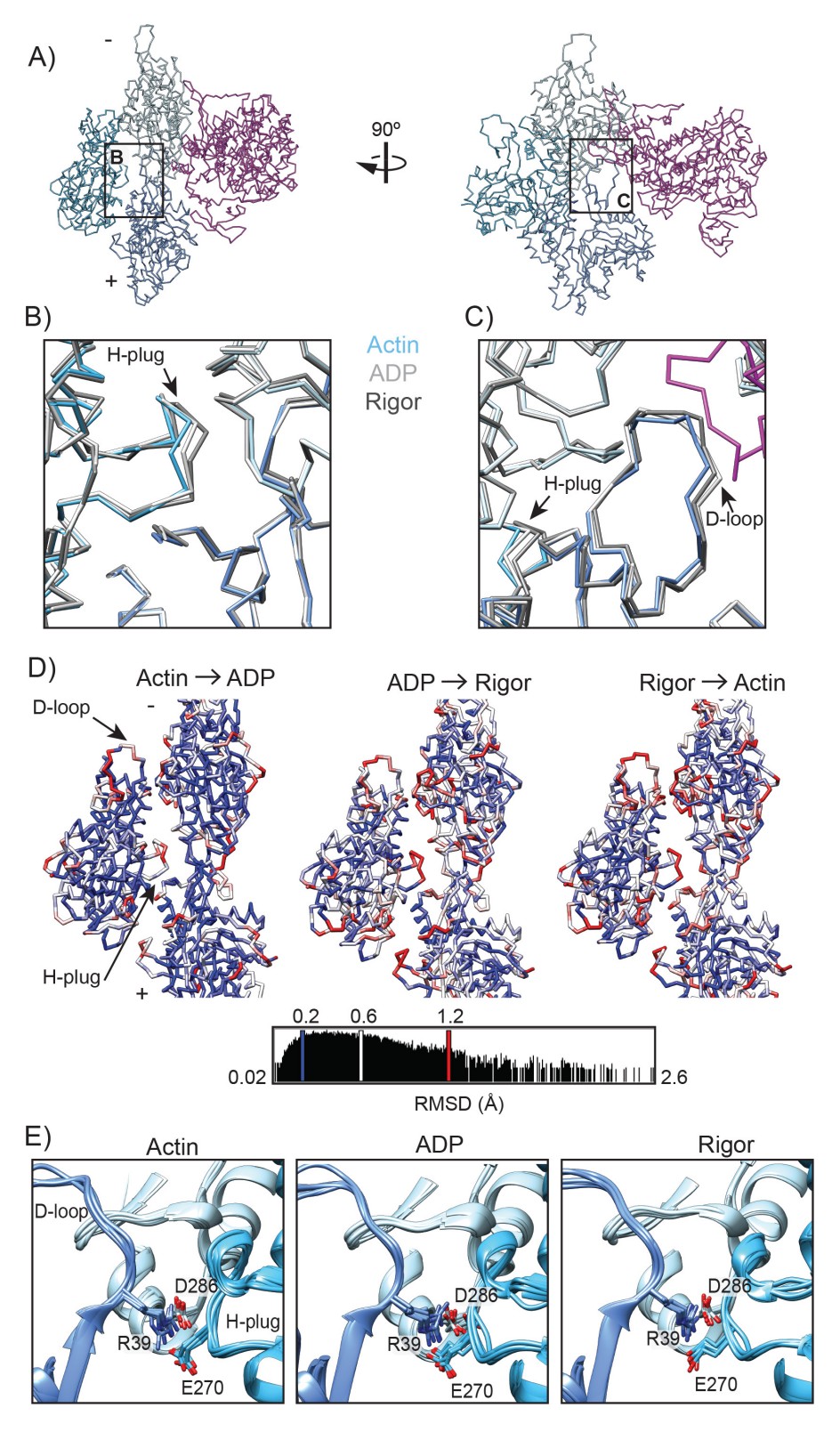

**Figure 6.** Actin rearrangements accompany force generation. (**A**) Backbone averaged trace of the HR MDFF rigor interface consisting of 1 MD (magenta) and three actin subunits (shades of blue). (**B**) Actin hydrophobic plug repositioning between actin alone (blue), ADP (light grey), and rigor (grey). To generate the displayed superposition, the full ADP and rigor density maps were aligned to the actin alone density map, and then the

*Figure 6 continued on next page*

*Figure 6 continued*

back-bone averaged HR MDFF model of each state was rigid-body fit into its corresponding density map. (**C**) View of D-loop displacements coupled to H-plug motion, colored and aligned as in B. Region of myosin contacting the D-loop from the rigor structure is displayed in magenta for reference. (**D**) Per-residue Cα RMSD is displayed between superpositions of backbone-averaged HR MDFF models, aligned as described in B. The backbone of the first state indicated is displayed and colored. Rearrangements of the largest magnitude occur in the D-loop and H-plug. (**E**) Superpositions of all six inter-strand interfaces from the indicated HR MDFF atomistic models (not averaged) displaying the interaction between D-loop R39 on actin protomer one with H-plug E270 on protomer two and D286 in SD3 of protomer 3. The interfaces were superimposed based on the Cα coordinates of actin protomer 1 (dark-blue) subunits, as described in *Figure 2A*. Colors are as in A.

DOI: https://doi.org/10.7554/eLife.31125.027

The following figure supplements are available for figure 6:

**Figure supplement 1.** EM density map comparison of H-plug and D-loop rearrangements between actin alone, ADP, and Rigor.

DOI: https://doi.org/10.7554/eLife.31125.028

**Figure supplement 2.** Differences in actin rearrangements between MyoVI and MyoIIC.

DOI: https://doi.org/10.7554/eLife.31125.029

---

subunits on the adjacent filament strand (*Chen et al., 1993*; *Holmes et al., 1990*). Unexpectedly, myosin VI engagement shifts the H-plug towards the two actins located on the opposite strand, with an increasing deviation of position from the unbound actin state as the motor proceeds from the ADP state (Cα RMSD 0.9 Å) to rigor (Cα RMSD 1.6 Å, *Figure 6B–D* and *Videos 6* and *7*).

Although the D-loop and H-plug are distal from one another within a single subunit, they are brought into close proximity between laterally adjacent protomers within the filament, and a single side-chain pair, D-loop residue R39 and H-plug residue E270, makes a direct contact between them across the interface (*Figure 6E*). The geometry is incompatible with these residues forming a canonical head-to-head interaction with their charged groups due to their close proximity; instead, they pack together through what may be a mixed electrostatic/Van der Waals interaction. R39 additionally forms a canonical head-to-head salt-bridge interaction with D286 of subdomain 3 of a longitudinally adjacent subunit, placing this residue at a vertex that connects three actin subunits along and across the strands of the filament (*Figure 6E*). We propose that this bi-partite interaction acts as an allosteric relay. As myosin VI remodels its binding site on actin, primarily through rearrangements of the D-loop, movement of the R39-D286 bridge necessitates repositioning of E270 due to steric exclusion. However, the electrostatic attraction between E270 and R39 prevents rotameric exchange of E270, producing instead a distortion of the H-plug.

## Discussion

Despite the fundamental conservation of enzymatic mechanism across the myosin superfamily, our studies unambiguously demonstrate that the actomyosin interface is a highly evolvable interaction surface. Comparing myosin IIC and myosin VI, we observe only a single identical actin-binding residue, myosin IIC P561/myosin VI P536 in the HLH; all other interfacial residues have diverged despite the high degree of conservation of the nucleotide-binding pocket (*Figure 7A*). The myosin VI interface residues we have identified are essentially completely conserved between species (*Figure 7A*), and likely contribute to this motor's specialization. The

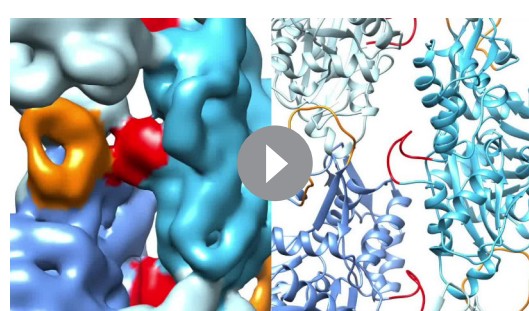

**Video 6.** Morph of the H-plug and D-loop between cryo-EM reconstructions of actin alone, ADP, and rigor, related to *Figure 6*. Morphs from actin alone to ADP, ADP to rigor, and rigor to actin alone focusing on the H-plug (red) and D-loop (orange). Right panels: Morphs of density maps, low-pass filtered to 7.5 Å and aligned to each other. Left panels: Morphs between backbone-averaged HR MDFF models. To generate the morphs of atomistic models, the density maps were aligned, then their corresponding atomistic models were rigid-body fit into the aligned maps.

DOI: https://doi.org/10.7554/eLife.31125.030

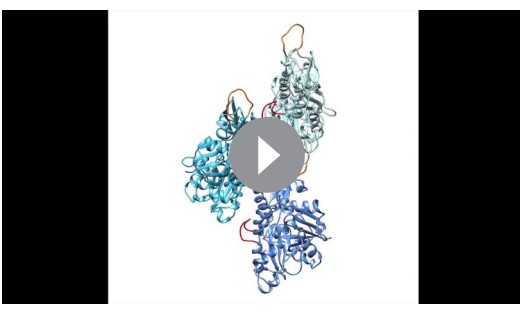

**Video 7.** Morph between 3 states of actin, related to *Figure 6*. Morph between the backbone-averaged HR MDFF actin alone, ADP, and rigor models. Morphs generated as in *Video 6*.
DOI: https://doi.org/10.7554/eLife.31125.031

most notable difference is the large increase in the number of loop 3 – actin contacts in myosin VI vs. IIC. This includes a proliferation of specific electrostatic interactions and the unanticipated myosin R561 – actin Y61 cation-π interaction which forms upon the transition from ADP to rigor in our models. These residues are poorly conserved among other myosin isoforms, suggesting that this interaction may have evolved to support the specialized properties of myosin VI. Future comparative structural studies of diverse myosins in complex with actin will be required for a detailed dissection of how modulation of the actomyosin interface correlates with motor-specific biophysical properties. Furthermore, our data suggest that the mechanisms of disease-causing interfacial mutations in other myosins will be difficult to predict from sequence analysis alone, and thus will likely require structural characterization.

Comparing our actin-bound reconstructions of the ADP and rigor states to previous structures of myosin VI in isolation in both pre- and post-power stroke states enables us to clarify the sequence of structural transitions which transduce ATP binding and hydrolysis into force production (*Video 8*). Nevertheless, the exact causal connections between nucleotide state, myosin conformation, and actin filament engagement remain to be fully resolved. Visualization of transient actin-binding interfaces in both pre-power stroke and post-rigor states that facilitate coordination of rearrangements in the nucleotide-binding cleft, actin-binding cleft, and converter will be necessary. The low-affinity of these high-energy states for actin renders this technically challenging; nevertheless, we optimistically anticipate that continued developments in cryo-EM methodology, including emerging methods for reconstructing filament – binding partner complexes with substoichiometric binding density (*Kim et al., 2016*; *Liu et al., 2017*), will render these as tractable structural targets in the future.

While the major lever arm swing from the PiR to the ADP state generates directional motion, we propose that the second, smaller lever arm bend upon the ADP to rigor transition contributes to the force sensitivity of the motor. We hypothesize that the directional strain on the myosin head can regulate the lever arm bending observed in our models during transitions between ADP and rigor (*Figure 5D*). Lever arm bending is coupled to converter rearrangements, which, through the SH1 and relay helices, promotes nucleotide cleft opening and ADP release (*Figures 5A, B* and *7B*). Rearward force applied to the lead head would disfavor the lever arm bend and thus promote the ADP-bound state; conversely, forward load would promote bending and thus favor the rigor state (*Figure 7B*). This framework is consistent with a model wherein extreme rearward force locks the motor in the ADP state, facilitating a transition from processive transporter to actin-bound tether (*Altman et al., 2004*; *Chuan et al., 2011*).

This proposal agrees with biochemical data demonstrating an increase in ADP dissociation rate under forward load and increased affinity for ADP under rearward load (*Altman et al., 2004*; *Oguchi et al., 2008*), and it is also consistent with recent simulation studies suggesting the converter can adopt a post-powerstroke conformation in the presence of force (*Mugnai and Thirumalai, 2017*). Kinetic measurements have suggested that ATP-binding is the force-sensitive step that coordinates the heads of a dimer via intramolecular strain (*Sweeney et al., 2007*); a structure of the post-rigor actomyosin VI transition complex would be required to develop a detailed mechanistic framework for force sensitivity in the rigor to post-rigor transition.

Mechanoregulated ADP release has also been reported for the processive dimeric transporter myosin V (*Oguchi et al., 2008*; *Wulf et al., 2016*), as well as the monomeric tethering motor myosin IC (*Greenberg et al., 2012*), and a small lever arm swing has also been reported for myosin V during this transition (*Veigel et al., 2002*). However, these motors each have a distinct structural topology linking the lever arm to the converter, with insert 2 of myosin VI, the site of lever arm bending, being a unique feature of myosin VI. Thus, further studies will be required to reveal mechanistic consonance and dissonance between these motors and themes in myosin force sensitivity. Additionally,

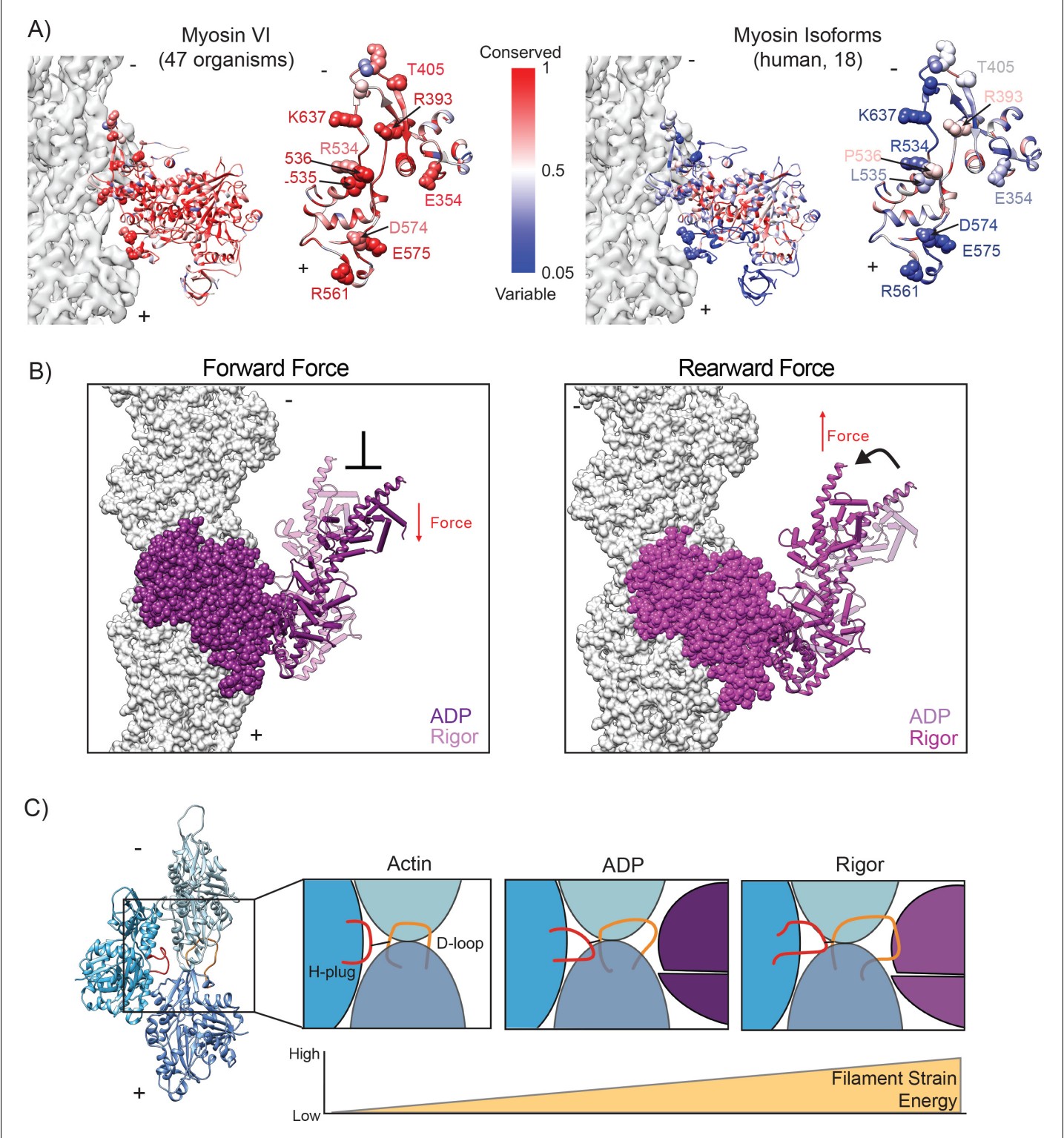

**Figure 7.** Conceptual models summarizing implications of the ADP to rigor transition. (**A**) Conservation of myosin VI between 47 organisms (left, Source Data 1) and among 18 human myosin isoforms (right, Source Data 2). Left, full MD; Right, en face view of actin binding interface with space-filling representation of critical residues mediating actin interaction. Actin density is displayed in transparent grey for reference. (**B**) Schematic of potential effects of force on the ADP to rigor transition. Due to the displacement associated with the lever arm bend, a rearward load should favor ADP engagement and a forward load should disfavor it. The displayed superposition was generated as in *Figure 5D*. (**C**) Cartoon depicting increased actin strain during myosin force generation. The MD (magenta) - D-loop (orange) interaction facilitates remodeling of the H-plug (red).

*Figure 7 continued on next page*

*Figure 7 continued*

DOI: https://doi.org/10.7554/eLife.31125.032

The following source data is available for figure 7:

**Source data 1.** FASTA sequences from NCBI BLAST search of 47 organisms used to determine conservation of myosin VI.

DOI: https://doi.org/10.7554/eLife.31125.033

**Source data 2.** FASTA sequences from NCBI BLAST search of 18 human myosin isoforms used to determine conservation of myosin VI.

DOI: https://doi.org/10.7554/eLife.31125.034

our high-resolution actomyosin VI structures provide a foundation for future motor engineering studies. Motor design represents a complementary route for investigating structural features conferring specific biophysical properties, including force sensitivity, which may additionally produce novel cytoskeletal motors with applications in biotechnology and biomedicine. Our studies suggest it may be feasible to alter stepping behaviors and force-sensitivity by exploring alternative lever-arm geometries which modulate bending and thereby the relative positioning of the lever-arm in ADP and rigor.

The structural transitions we observe in actin may play an important role in the actomyosin VI force generation cycle. The anomalously high level of conservation in actin has been ascribed to a requirement for allosteric coordination between subunits (*Galkin et al., 2010*; *Galkin et al., 2015*), as we indeed observe, leading us to speculate at potential functional roles for these rearrangements. As the unbound actin state represents the conformation adopted by F-actin in the absence of exogenous factors, we propose that our free actin model represents a low-energy conformation of the H-plug and D loop in the context of the filament. As we observe the conformation of the H-plug becomes increasingly deformed as the motor proceeds from ADP to rigor, with increasing RMSD relative to the ground-state of unbound actin (*Figure 6D*), we speculate that this segment is adopting an increasingly unfavorable conformation as the binding affinity of the actomyosin interface increases, suggesting strain energy may be stored in the filament as the force generation cycle proceeds (*Figure 7A*). We propose that the D-loop acts as a 'handle' which enables myosin VI, and potentially other actin-binding proteins, to transmit conformational changes from the filament surface through an allosteric relay to the H-plug (*Figure 7A*), converting binding energy into strain energy. As our data suggest that the filament is maximally strained when the motor is most stably bound in rigor, one possibility we envision is that strain energy stored in the filament facilitates MD displacement as ATP rebinds during the transition to post-rigor, with elastic recoil in the actin filament helping to drive this transition forward (*Figure 7A*). Examination of the actomyosin IIC rigor complex suggests that D-loop remodeling by this motor does not produce H-plug distortion (*Figure 6—figure supplement 2*), indicating that this mechanism may be selectively employed by different myosins and represents another avenue for motor specialization. This model is consistent with previous functional data

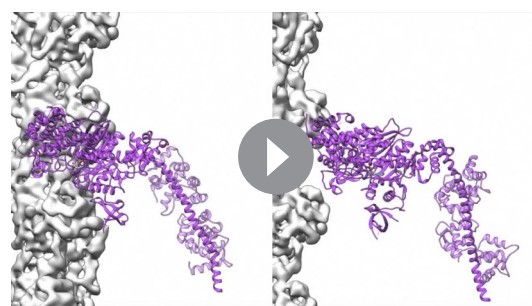

**Video 8.** The myosin VI mechanochemical cycle. Morph between indicated states of the myosin VI mechanochemical cycle, aligned in the reference from of the actin filament. To generate this morph, the full ADP and rigor density maps were aligned, and then the LPF ADP MDFF and LPF rigor MDFF models were rigid body fit into their corresponding densities. The grafted lever arm models were then superimposed on the LPF MDFF models using overlapping Cα coordinates. For the Pre-Powerstroke (2V26) and PiR (4PFO) states, the X-ray structures were superimposed on the LPF ADP MDFF model based on the Cα coordinates of the full motor domain. For the post-rigor (2VAS) state, the x-ray structure was superimposed on the LPF rigor MDFF model based on the Cα coordinates of the full motor domain. The lever arms of crystal structures were extended in a manner analogous to the procedure described in the Experimental Methods after superposing 3GN4 utilizing overlapping Cα coordinates. Myosin, magenta, actin density map, grey. Nucleotides and ions are displayed in ball and stick representation and colored by heteroatom.

DOI: https://doi.org/10.7554/eLife.31125.035

suggesting differential F-actin conformational dynamics in the presence of myosin V and muscle myosin S1 (*Prochniewicz et al., 2010*)

An additional and non-exclusive possibility is that myosin VI-induced actin conformational states are also modulated through additional mechanisms to regulate the activity of the motor in a context-dependent manner. Actin nucleotide state, a marker of filament age within the cell, influences myosin VI processivity (*Zimmermann et al., 2015*), and could exert its effects via such a mechanism. Finally, conformational changes generated at a single actomyosin VI interface could be allosterically communicated along a filament to influence the binding interactions or activity of other actin binding partners at distal sites, consistent with previous reports of myosin VI influencing the structural dynamics and mechanical rigidity of actin (*Prochniewicz et al., 2011*). While our structural data clearly suggest that conformational changes should propagated across the lateral interface between strands, the technical necessity of saturating the filaments with myosin for high-resolution reconstruction is not compatible with visualizing rearrangements induced at a distance. Future structural studies of myosin VI bound to F-actin trapped in different nucleotide states, as well as filaments sparsely decorated with this motor, will facilitate experimental testing of these proposals.

## Materials and methods

### Buffers
KMEI: 50 mM KCl, 1 mM $MgCl_2$, 1 mM EGTA, 10 mM imidazole, pH 7.0. G-Mg: 2 mM Tris, pH 8, 0.5 mM DTT, 0.2 M ATP, 0.1 mM $MgCl_2$, 0.01% $NaN_3$.

### Protein expression and purification
Myosin VI was engineered and purified as previously described (*Omabegho et al., 2017*). Briefly, a DNA construct for protein expression was assembled from fragments encoding porcine myosin VI (residues 1–817) and *Archaeoglobus fulgidus* L7Ae (residues 9–118), cloned into a pBiex-1 (Novagen-Millipore, Burlington, MA) expression vector modified to include codons for a C-terminal eYFP, and FLAG tag (DYKDDDDK) with intervening GSG repeats (see *Figure 1—figure supplement 1*). Proteins were expressed by direct transfection of SF9 cells and affinity purified as previously described (*Elting et al., 2011*; *Liao et al., 2009*). Rabbit skeletal muscle actin was prepared as previously described (*Pardee and Spudich, 1982*). F-actin was prepared by polymerizing 10 µM actin monomers in KMEI + G-Mg buffer overnight at 4°C.

### Cryo-EM sample preparation
F-actin and myosin constructs were diluted to 0.3–0.6 µM and 2–4 µM, respectively, in KMEI. For nucleotide free conditions, myosin samples were supplemented with 10 U/mL apyrase (Sigma); for ADP conditions, myosin samples were supplemented with 5 mM Mg-ADP (Sigma-Aldrich, St. Louis, MO) pH 7.0. F-actin (3 µL) was applied to a plasma-cleaned 1.2/1.3 200-mesh C-flat holey carbon grid (Protochips, Morrisville, NC) in the humidified chamber of a Leica GP plunge freezer and incubated for 60 s at 25°C. Myosin (3 µL) was then applied and incubated for 60 s. Solution (3 µL) was then removed and an additional 3 µL of myosin was applied. After an additional 60 s, 3 µL of solution was removed, and then the grid was blotted for 2–3 s from the back with filter paper (Whatman no 5.) and plunge-frozen in liquid ethane.

### Cryo-EM data collection
Cryo-EM data were collected with the Leginon software on a Tecnai F20 operating at 200 kV using a Gatan K2 Summit direct electron detector in counting mode. Movies were collected with an exposure of 0.25 s/frame for a total of 6.0 s (24 frames) at a dosage of 6 e⁻/Å²/s (7.6 e⁻/pixel/s) yielding a total cumulative dose of 36 e⁻/Å². Data were collected at 1.5–3 µm underfocus at a nominal magnification of 29,000x, corresponding to a calibrated pixel size of 1.27 Å at the specimen level.

### Image processing
For initial processing steps, image frames were aligned and summed with Unblur (*Grant and Grigorieff, 2015*) without dose-weighting. Contrast transfer function (CTF) estimation and extraction of segments was performed in the Appion data-processing environment (*Lander et al., 2009*). Unless

otherwise specified, 2D image processing operations were carried out using proc2d from the EMAN processing package (*Ludtke et al., 1999*). CTF parameters were estimated with CTFFIND3 (*Mindell and Grigorieff, 2003*). Segments were windowed in 512-pixel boxes with 81 Å of non-over-lap corresponding to a step-size of 3 actin protomers, normalized with xmipp_normalize (*Scheres et al., 2008*), then binned by 2. Segments were extracted for each state: ADP (36,114), rigor (56,116), and actin alone (63,139).

For 3D refinement and reconstruction, we adapted the IHRSR protocol recently described in Kim et al. (*Kim et al., 2016*), performing initial refinement and reconstruction using functions from the SPARX/EMAN2 (*Hohn et al., 2007*; *Tang et al., 2007*) libraries and helical search using the program hsearch_lorentz (*Egelman, 2007*), followed by final refinement and reconstruction using FREALIGN (*Lyumkis et al., 2013*). Briefly, segments were extracted from phase-flipped images, then refined against an initial model generated by low-pass filtering an actin reconstruction (EMD-1990 [*Behrmann et al., 2012*]) to 35 Å. The reconstruction obtained from this refinement run was then low-pass filtered to 35 Å and used as the initial model for a second round of refinement, where poorly aligning segments were excluded using a cross-correlation cutoff of 1.5 σ. Segments with cor-relation scores above the cutoff were then divided into two random half-datasets, and independent refinement of these half-datasets (to minimize noise bias [*Scheres and Chen, 2012*]) was re-initial-ized using the same low-pass filtered initial model. After each round of refinement, the asymmetric reconstructions of the half-datasets were summed, and the sum was used to calculate new helical parameters. These helical parameters were then applied to each half-reconstruction independently, which were then compared and low-pass filtered based on the Fourier Shell Correlation (FSC) to pro-vide the references for the next round of refinement.

After refinement in EMAN2/SPARX, un-binned segments were generated using alignparts_lmbfgs (*Rubinstein and Brubaker, 2015*) on all acquired frames to correct for non-uniform beam-induced drift (motion correction) and apply an exposure-dependent filter to maximize signal at all spatial fre-quencies (*Grant and Grigorieff, 2015*). Data from all frames were included in segments extracted at this stage. Parameters from the half-data sets were recombined, then final refinement and recon-struction was performed with FREALIGN v 9.11 using fixed helical parameters and a strict low-pass filter of 10 Å, as we found including higher-resolution information in the refinement did not improve the reconstructions (data not shown). The final average resolutions reported were determined based on the FSC 0.143 criterion (*Rosenthal and Henderson, 2003*) as 4.6 Å (rigor), 5.5 Å (ADP), and 5 Å (actin alone) (*Figure 1—figure supplement 1*). The maps were sharpened using a B-factor peaking at the nominal average resolution as indicated in *Table 1* using the program BFACTOR.

Local resolution assessment was performed in two independent fashions. Since it was clear that the resolution decayed radially from the core of the filament, we calculated a series of reconstruc-tions with cylindrical masks of radii chosen to exclude certain portions of the map: 120 Å radius for the full map, 90 Å radius to exclude the converter and lever arm (which was used to calculate the overall resolution reported), and 40 Å radius for the actomyosin interface (*Figure 1—figure supple-ment 1*). Resolutions were determined for each individual reconstruction based on the FSC 0.143 cri-terion. Local resolution was also estimated for strong-bound ADP and rigor states using ResMap (*Kucukelbir et al., 2014*) on the full density maps revealing a resolution gradient of better than ~4 Å (actin) to worse than 5–6 Å (myosin lever arm).

## Building atomic models with MDFF

Atomistic models for the cryo-EM density maps were generated using the Molecular Dynamics Flexi-ble Fitting (MDFF) procedure. Initial models were built from eight actin subunits (3J8A) and six myo-sins (2BKI for rigor state, 4PFO ADP-strongbound). Two models were generated for each state: A high resolution (HR) model for actomyosin interface analysis and a low pass filtered (LPF) model for analysis of global MD rearrangements. For the HR model, the MD was truncated to exclude the lever arm and converter regions, and the electron density maps used were B-factor sharpened and filtered to nominal resolution as indicated in *Table 1*. As there is no pre-existing structural information for the flexible loops, loop 2 and the HCM loop, we manually constructed these regions using Coot (*Emsley et al., 2010*). Initial models were then assembled through rigid body docking in Chimera, followed by flexible fitting with DIREX (*Wang and Schröder, 2012*).

MDFF was performed with explicit solvent, 50 mM KCl, and symmetry restraints imposed on Cα of actin. The simulation was run in three steps: a brief energy minimization step to remove severe

clashes from the starting model, then molecular dynamics with low map weighting (250ns simulation), followed by a longer energy minimization (2000 steps) using a higher map-weighting. To accommodate the multi resolution maps and the different resolutions of each density map, each state was subjected to MDFF differently: Due to the higher resolution of the rigor state density map, backbone atoms and large side chain atoms in actin (Phe, Tyr, Trp, His, Arg, Gln, Lys, Met) were subjected to fitting by the electron density map potential. For all other models (ADP, actin alone, and 'low resolution' models), only backbone atoms were permitted to feel map potential. Loop two and the HCM loop were excluded from flexible fitting and only subjected to molecular dynamics, and the positions of ADP and magnesium ions were kept fixed during the molecular dynamics simulation due to the limited resolution of the reconstructions. We tested various values of the weighting factor 'g' for both the molecular dynamics and the long energy minimization stages and selected the optimal value by assessing quality using Molprobity (*Chen et al., 2010*) as implemented in Phenix (*Adams et al., 2010*) as described previously (*Kim et al., 2016*).

To generate the LPF models, for both ADP and rigor state, the lever arm and converter (and CaM for the rigor structure) from 2BKI were grafted onto the HR atomistic models, and electron density maps were B-factor sharpened and filtered to 7.5 Å resolution to accommodate the lower resolution portions of the map consisting of the converter and lever arm. Initial fitting was carried out through rigid body docking in Chimera and then flexible fitting with Direx. The MDFF was carried out in the same manner as the 'high resolution' models with the filtered density maps guiding fitting for only the protein main-chain atoms. Due to the overall lower resolution of the filtered maps, atomistic models were backbone averaged in Phenix (*Adams et al., 2010*) and side chains were truncated to poly-alanine. An analogous backbone averaging procedure was applied to actin subunits from the HR models to visualize conformational changes in actin between states in the force generation cycle (*Figure 6*).

To create the extended lever arm models, the lever arm and 2 calmodulins from 3GN4 (truncated at K848) were fit as a rigid body into the 7.5 Å resolution filtered density maps, which were segmented to only include the motor domain, converter, calmodulin and ordered region of the lever arm. After fitting, the lever arm was truncated, then grafted onto the LPF model at a site chosen to match the local path of the density. As the fits were dependent on the correspondence between the first IQ and calmodulin from the crystal structure and our density maps, only the lever arm and one calmodulin were fit into the density; thus, the second calmodulin and extension of the lever arm are extrapolations based on the crystal structure. The lever arm was grafted at V784 for the rigor model and D773 for the ADP model.

## Conservation analysis

Conservation analysis was carried out through sequence alignment using the EMBL-EBI Clustal Omega server (*Sievers et al., 2011*) of human myosin VI sequence with myosin VI sequences from 46 other organisms or 18 other human myosin isoforms obtained through a NCBI BLAST search. Conservation mapping onto the myosin VI structure was conducted in Chimera.

## Quantification and statistical analysis

Density map alignments, structural superpositions of atomistic models, RMSD calculations, centroid determinations, and displacement calculations were conducted in UCSF Chimera (*Pettersen et al., 2004*). Inter-domain rotation axes and angles were calculated using DynDom3d (*Poornam et al., 2009*). Cα displacement vectors were calculated using a Python script which has previously been described (*Alushin et al., 2014*).

## Data and software availability

Cryo-EM density maps and corresponding atomistic models for rigor, ADP, and actin alone reconstructions have been deposited in the Electron Microscopy Data Bank (EMDB) and Protein Data Bank (PDB). Electron Microscopy Data Bank accession codes: EMD-7115 (actin alone), EMD-7116 (rigor), EMD-7117 (ADP). Protein Data Bank accession codes: Actin alone: 6BNO (HR MDFF), 6BNU (averaged HR MDFF); Rigor: 6BNP (HR MDFF), 6BNV (LPF MDFF); ADP: 6BNQ (HR MDFF), 6BNW (LPF MDFF). All custom software utilized in structure determination and analysis are available at:

https://github.com/alushinlab/goldhelix (*Alushin, 2017*; copy archived at https://github.com/elifes-ciences-publications/goldhelix).

## Acknowledgements

We gratefully acknowledge Jenny Hinshaw (NIDDK) for sharing equipment and microscope usage. We also thank Jim Sellers and Harry Takagi (NHLBI) for thoughtful comments and discussion. This work was supported by a Women and Science Fellowship from the Rockefeller University to PSG, an NIH Fellowship (F32GM094420) to TO, and a Human Frontiers Science Program Long-Term Fellowship to PVR. Additional funding was provided by the Division of Intramural Research of the National Heart, Lung, and Blood Institute to GMA, a grant from the WM Keck Foundation to ZB, and NIH High-Risk High-Reward Research Grants 1DP2 OD004690 to ZB and 5DP5OD017885 to GMA.

## Additional information

### Funding

| Funder | Grant reference number | Author |
|---|---|---|
| W. M. Keck Foundation | | Zev Bryant |
| Human Frontier Science Program | Long-Term Fellowship | Paul V Ruijgrok |
| National Heart, Lung, and Blood Institute | | Gregory M Alushin |
| Rockefeller University | Women & Science Fellowship | Pinar S Gurel |
| National Institutes of Health | F32GM094420 | Tosan Omabegho |
| National Institutes of Health | 1DP2 OD004690 | Zev Bryant |
| National Institutes of Health | 5DP5OD017885 | Gregory M Alushin |

The funders had no role in study design, data collection and interpretation, or the decision to submit the work for publication.

### Author contributions

Pinar S Gurel, Conceptualization, Data curation, Formal analysis, Investigation, Methodology, Writing—original draft, Writing—review and editing; Laura Y Kim, Data curation, Investigation, Writing—review and editing; Paul V Ruijgrok, Resources, Data curation, Writing—review and editing; Tosan Omabegho, Resources, Data curation; Zev Bryant, Conceptualization, Funding acquisition, Validation, Project administration, Writing—review and editing; Gregory M Alushin, Conceptualization, Software, Supervision, Funding acquisition, Validation, Investigation, Visualization, Writing—original draft, Project administration, Writing—review and editing

### Author ORCIDs

Gregory M Alushin http://orcid.org/0000-0001-7250-4484

### Decision letter and Author response

Decision letter https://doi.org/10.7554/eLife.31125.057
Author response https://doi.org/10.7554/eLife.31125.058

## Additional files

### Supplementary files

• Transparent reporting form
DOI: https://doi.org/10.7554/eLife.31125.036

## Major datasets

The following datasets were generated:

| Author(s) | Year | Dataset title | Dataset URL | Database, license, and accessibility information |
|---|---|---|---|---|
| Pinar S Gurel, Gregory M Alushin | 2017 | Structure of bare actin filament | http://emsearch.rutgers.edu/atlas/7115_summary.html | Publicly available at the EMDataBank (accession no. EMD-7115) |
| Pinar S Gurel, Gregory M Alushin | 2017 | CryoEM structure of MyosinVI-actin complex in the rigor (nucleotide-free) state | http://emsearch.rutgers.edu/atlas/7116_summary.html | Publicly available at the EMDataBank (accession no. EMD-7116) |
| Pinar S Gurel, Gregory M Alushin | 2017 | CryoEM structure of Myosin VI-Actin complex in the ADP state | http://emsearch.rutgers.edu/atlas/7117_summary.html | Publicly available at the EMDataBank (accession no. EMD-7117) |
| Pinar S Gurel, Gregory M Alushin | 2017 | Structure of bare actin filament, backbone-averaged with sidechains truncated to alanine | https://www.rcsb.org/pdb/explore/explore.do?structureId=6BNU | Publicly available at the RCSB Protein Data Bank (accession no. 6BNU) |
| Pinar S Gurel, Gregory M Alushin | 2017 | Structure of bare actin filament | https://www.rcsb.org/pdb/explore/explore.do?structureId=6BNO | Publicly available at the RCSB Protein Data Bank (accession no. 6BNO) |
| Pinar S Gurel, Gregory M Alushin | 2017 | CryoEM structure of MyosinVI-actin complex in the rigor (nucleotide-free) state | https://www.rcsb.org/pdb/explore/explore.do?structureId=6BNP | Publicly available at the RCSB Protein Data Bank (accession no. 6BNP) |
| Pinar S Gurel, Gregory M Alushin | 2017 | CryoEM structure of MyosinVI-actin complex in the rigor (nucleotide-free) state, backbone-averaged with side chains truncated to alanine | https://www.rcsb.org/pdb/explore/explore.do?structureId=6BNV | Publicly available at the RCSB Protein Data Bank (accession no. 6BNV) |
| Pinar S Gurel, Gregory M Alushin | 2017 | CryoEM structure of Myosin VI-Actin complex in the ADP state | https://www.rcsb.org/pdb/explore/explore.do?structureId=6BNQ | Publicly available at the RCSB Protein Data Bank (accession no. 6BNQ) |
| Pinar S Gurel, Gregory M Alushin | 2017 | CryoEM structure of Myosin VI-Actin complex in the ADP state, backbone-averaged with side chains truncated to alanine | https://www.rcsb.org/pdb/explore/explore.do?structureId=6BNW | Publicly available at the RCSB Protein Data Bank (accession no. 6BNW) |

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
