## [Decision Letter]

Thank you for submitting your article "Cryo-EM structures reveal specialization at the myosin VI-actin interface and a mechanism of force sensitivity" for consideration by *eLife*. Your article has been favorably evaluated by Andrea Musacchio (Senior Editor) and three reviewers, one of whom is a member of our Board of Reviewing Editors. The following individual involved in review of your submission has agreed to reveal their identity: Anne Houdusse (Reviewer #3).

The reviewers have discussed the reviews with one another and the Reviewing Editor has drafted this decision to help you prepare a revised submission.

Summary:

This is a very nice study, presenting a comparison of Cryo-EM structures of myosin VI bound to actin under rigor conditions (free of nucleotides), and in the presence of MgADP. The observed changes between myosins bound to actin in these two states are novel. They lead the authors to propose an interesting mechanosensitivity role for the ADP-to-rigor structural transition, speculating about a connection between lever arm bending and the nucleotide cleft state (in +/- ADP) in myosin VI. One of the interesting conclusions when a comparison is made with other myosins is the extent to which myosin residues at the actin interface have not been conserved. Another interesting conclusion involves the extent of actin conformational changes, particularly those distant from the myosin-binding sites.

Essential revisions:

1) Reviewers were surprised by the statement in the Abstract: "suggesting a previously unanticipated role for F-actin structural plasticity during force generation." This would only be surprising if one were unaware of a fairly extensive literature on this topic. Consider Drummond et al., "Alteration in crossbridge kinetics caused by mutations in actin", Nature (1990). They stated "One of the mutations is not in the putative myosin-binding site, demonstrating the importance of long-range effects of amino acids on actin function" and "amino acid 336… is distant from the nearest known myosin or tropomyosin contact… suggesting that E316K may be affecting the interaction with myosin through a conformational change in the actin." Current high-resolution models of actin-tropomyosin-myosin are consistent with the lack of any such possible direct interaction. Sparrow and colleagues went on (in Anson et al., 1995) to show that the E316K mutation has no change in rigor affinity for myosin S1 but a greatly reduced in vitro motility.

Or consider Prochniewicz and Yanagida (JMB, 1990), "Inhibition of sliding movement of F-actin by crosslinking emphasizes the role of actin structure in the mechanism of motility". Or Prochniewicz et al. (Biophysical Journal, 1993) where they state: "These results suggest that the interaction of actin with myosin involves cooperative structural changes in F-actin propagated through intermonomer bonds along the filament." Or Schwyter et al., (JCB, 1990): "These results suggest that the motility of actin filaments can be uncoupled from the activation of myosin ATPase activity and is dependent on the structural integrity of actin and perhaps, dynamic changes in the actin molecule." Or Kim et al. (Biochemistry, 1998): "These results show the uncoupling between force generation and other events in actomyosin interactions and emphasize the role of actin filament structure and dynamics in the contractile process." A number of these observations are summarized in a Minireview (Galkin et al., Current Biology, 2012): "But a number of papers have shown that actin can be modified, either chemically [27-30], by mutation [31] or by proteolysis [32], in a way that inhibits myosin force generation without inhibiting either the binding of myosin to actin or the actin-induced activation of myosin's ATPase activity. The simplest explanation for these observations is that actin must undergo structural transitions during actomyosin force generation, and these modifications of actin inhibit such structural transitions."

Also, one might consider the cooperativity of such changes, as in Orlova and Egelman, "Cooperative rigor binding of myosin to actin is a function of F-actin structure", (JMB, 1997): "Thus, while the cooperative binding of myosin to actin during muscle activation has been widely studied, we think that the results reported here suggest that one component of this cooperativity is within actin itself." This has been extended by several papers from Uyeda, which discuss myosin-induced cooperative conformational changes in actin.

There are also Reisler's observations on intermolecular dynamics in actin filaments (Kim and Reisler; Biophys. Chem. 2000), and on cofilin induced cooperative effects on actin structure and dynamics as documented by AFM in Uyeda work (and by Bobkov et al., JMB 2006) can be included in this context. It is noteworthy that the title of Ngo et al. is "Allosteric regulation by cooperative conformational changes of actin filaments drives mutually exclusive binding with cofilin and myosin".

The authors should also look at the pyrene quenching data in F-actin. Nucleotide effects in actomyosin-II were reported with probes (PMID: 15311925), and the Wulf et al. 2016 paper also described a change in F-actin parameters depending on whether ADP or Rigor conditions are used.

We would therefore suggest that a revision be made, replacing "suggesting a previously unanticipated role for F-actin structural plasticity during force generation" with "supporting many previous observations and suggestions about an important role for F-actin structural plasticity during force generation".

2) There were substantial concerns that some results are over-interpreted given the relatively modest resolution (by the new standards in cryo-EM). For example: "The Cα RMSD between the U50 and L50 from the rigor and ADP states is 1.467 Å and 1.118 Å". This appears silly. There is a resolution in this region of ~ 4.7 Å. Do they really expect that they can determine rms deviations to a precision of 1/1000 of an Å? All such rmsd values in the paper should probably be rounded to the nearest integer. Or: "The cation-pi interaction between R561 in loop 3 with Y91 of actin SD2 is not present for the ADP state, suggesting that this contact forms upon the transition from ADP to rigor (Figure 2)." But Figure 2 is a model. Do the authors think that the density map shows this directly? If so, this should be shown. If not, such statements in the paper need to be advanced with much more caution and with the proper caveats.

It is critical that the authors do a better job in describing and illustrating the EM map rather than its interpretation whenever they discuss small differences that would be critical for motor function. By showing only the model built into the modest resolution map, there is no way the reader can assess whether the differences come from various possible models that can be built or whether they are intrinsic to a well-defined change in the structure of the ADP strong state. There is no figure and description of the nucleotide-binding site to ascertain of the presence (StrongADP) or absence (Rigor) of the nucleotide in these structures. Similar for the transducer for which it is not clear whether the maps have the resolution to support a change in the transducer and how large this change is. The only density map shown is that of the nucleotide binding site of the actin filament, which is a good part of the map although there is clearly lack of precision to position the Calpha and side chain unambiguously at this resolution. The resolution of these reconstructions are not as high as those reported for NM2C.

The current figures do not provide access to the EM data itself and its interpretation for the converter, the relay, the SH1 helix and Switch II whose conformations are described in detail as undergoing important changes during the transition between Strong ADP and Rigor.

The authors build a longer lever arm (including the IQ motif) from an Xray structure while this is not present in their map (since the sample studied is a chimera) – how good is the density to properly orient the converter and to describe its conformation? This should be presented in a zoom of this region (movie), directly comparing the position of the density in rigor versus strong ADP.

The elongation of a particular orientation via addition of the rest of the lever arm magnifies the changes which seem overall rather small in the region of the converter and inser-2/Cam and which also suffer from lack of continuous density. There is no direct data supporting the orientation of the lever arm after the Cam bound to insert-2 for which the density is rather poor – thus the angle measurements provided seem speculative, or should be carefully presented.

Note also that the reference to Lister et al. 2004 to validate the change between ADP and Rigor is not appropriate since the molecule studied under optical trap is the full myosin VI in which the lever arm is much longer than what is presented in Figure 4 – thus the 10 A coincidence between the model shown in Figure 4 and the optical trap data (Lister et al) is not appropriate and should be removed.

3) The previously reported ADP release related lever arm swing in myosin V, which doesn't contain the insert 2 region, calls for greater caution in the interpretation of present results.

4) It is also puzzling that a lot of previous publications on the role of specific sites on actin in the binding of myosin (+/- nucleotides), as examined by mutational and structural approaches (from Rubenstein, Thomas, Reisler, and other groups), including the publication on hydroxyl radical mapping (footprinting) of myosin II sites on actin (Otzug-Durer et al.) are not considered here. Because the authors don't expect significant variations in actin binding sites for different myosins, this weakness should be corrected in the revised manuscript.

5) What differs between Strong ADP and Rigor are the main new data presented in this paper so it should be the main focus of the result presentation, avoiding speculative statements and clearly describing these differences and their link to the presence or absence of ADP in the molecule.

The authors should rewrite the comparison between Strong ADP and Rigor and be conservative about what is clearly demonstrated by their new EM data and what comes from a speculation pushing its interpretation. When it comes to speculation, this should be described in Discussion rather than the Results and more caution should be clearly stated.

To validate the model of the Strong ADP map interpretation, since it is of lower resolution, it is critical to compare different models one would obtain upon starting from different myosin VI models (post-rigor or rigor-like, in addition to the start as PiR already presented). Whether the final models after fitting would all converge in critical regions that differ between the Strong ADP and Rigor states or whether they would end up showing different interpretations need to be described. Regions that vary should indicate either a lack of restraint from the EM map, which could be linked to variability of the region, or just lack of resolution of the current data. It would be important to describe whether there is sufficient information to place the Calpha (and/or side chain) unambiguously whenever the authors use the differences to suggest consequences in motor function (strong ADP/Rigor transition and actin structure changes during the powerstroke). The detailed descriptions at atomic level (actin interface, in particular for the Milligan cation-pi interaction that would differ in Rigor and Strong ADP states should only be included if the EM density gives more confidence that the data support such atomic descriptions.

6) How can the authors guarantee that the Strong ADP data used correspond to mainly one state despite the lack of resolution in particular for the regions that are distal (relay / SH1 helix / lever arm)? What limits the resolution of the current studies? More detail should be provided about motion correction and the selection of frames used to calculate the EM maps. How can the authors ascertain that the lack of resolution for the Strong ADP is not linked to population of different conformers which should thus raise caution in the interpretation of the map? The density of the converter and the insert2/Cam allowing the building of the model is poorly described as well as the limitations that result from it. The authors should be careful in avoiding over-interpretation based on careful description of the quality and validation of the model in different parts of the molecule. In this sense, they should also describe more carefully whether there are changes at the actin/myosin interface that would be associated also with this transition. If they don't have the resolution to say so, (since the HCM loop and loop2 density is missing from their map), they should state that higher resolution of the reconstruction is likely required to access these important details of the actin/myosin interface.

---

## [Author Response]

Essential revisions:1) […] We would therefore suggest that a revision be made, replacing "suggesting a previously unanticipated role for F-actin structural plasticity during force generation" with "supporting many previous observations and suggestions about an important role for F-actin structural plasticity during force generation".

We thank the reviewers for this suggestion and the very detailed review of relevant literature. We have revised our Abstract in accordance with the reviewers’ suggestion, clarifying that the specific new contribution of our study in this area is direct visualization of structural changes in actin during force generation by myosin VI. Additionally, we have expanded our Introduction to include discussion of previous literature suggesting a role for actin structural plasticity in force generation (last paragraph), and have incorporated several of the references the reviewers suggested at other locations throughout the manuscript.

2) There were substantial concerns that some results are over-interpreted given the relatively modest resolution (by the new standards in cryo-EM). For example: "The Cα RMSD between the U50 and L50 from the rigor and ADP states is 1.467 Å and 1.118 Å". This appears silly. There is a resolution in this region of ~ 4.7 Å. Do they really expect that they can determine rms deviations to a precision of 1/1000 of an Å? All such rmsd values in the paper should probably be rounded to the nearest integer.

The reviewers’ point in this regard is well-taken. We have revised the manuscript so that all RMSD values are rounded to a single decimal place.

Or: "The cation-pi interaction between R561 in loop 3 with Y91 of actin SD2 is not present for the ADP state, suggesting that this contact forms upon the transition from ADP to rigor (Figure 2)." But Figure 2 is a model. Do the authors think that the density map shows this directly? If so, this should be shown. If not, such statements in the paper need to be advanced with much more caution and with the proper caveats.

We agree with the reviewers that caution must be employed when claiming side-chain level detail at our resolution, which we have clarified at appropriate locations in the text (Results section). We have revised the language in the referenced paragraph, and also now provide the corresponding density maps for this figure, which do demonstrate clear density for this contact in the rigor state (visible due to the large size of the side-chains involved), but absent in the ADP state. In the revised manuscript, this figure is now Figure 3.

It is critical that the authors do a better job in describing and illustrating the EM map rather than its interpretation whenever they discuss small differences that would be critical for motor function. By showing only the model built into the modest resolution map, there is no way the reader can assess whether the differences come from various possible models that can be built or whether they are intrinsic to a well-defined change in the structure of the ADP strong state.

We now systematically provide images of the EM density maps in order to better interpret the differences between ADP and rigor (Figure 2, Figure 3, Figure 3—figure supplement 1, Figure 3—figure supplement 3, Figure 3—figure supplement 5, Figure 4, Figure 5, Figure 5—figure supplement 1, Figure 5—figure supplement 2, and Figure 6—figure supplement 1).

There is no figure and description of the nucleotide-binding site to ascertain of the presence (StrongADP) or absence (Rigor) of the nucleotide in these structures.

In Figure 3—figure supplement 1 and Video 5, we now show the nucleotide binding cleft of myosin VI, as the reviewers have requested, and observe clear density in the ADP map that is notably absent in the rigor map. Thus, there is a clear density peak for the nucleotide, supporting our claim that we have visualized an ADP state. However, it is not possible to build a detailed model of a protein-ligand interaction at our resolution. We have therefore clarified in our figure legends that the ADP displayed in the myosin binding pocket is a visual guide; we do not intend to make any claims about the detailed stereochemistry of the interaction.

Similar for the transducer for which it is not clear whether the maps have the resolution to support a change in the transducer and how large this change is.

We have also now included density for the transducer in Figure 3—figure supplement 5, to directly show the data supporting the rearrangements we describe in the text.

The only density map shown is that of the nucleotide binding site of the actin filament, which is a good part of the map although there is clearly lack of precision to position the Calpha and side chain unambiguously at this resolution. The resolution of these reconstructions are not as high as those reported for NM2C.The current figures do not provide access to the EM data itself and its interpretation for the converter, the relay, the SH1 helix and Switch II whose conformations are described in detail as undergoing important changes during the transition between Strong ADP and Rigor.

We now provide EM density maps for all of the following motor regions: Converter (Figure 4—figure supplement 1), relay, SH1 helix (Figure 4—figure supplement 2), and Switch II (Figure 5).

The authors build a longer lever arm (including the IQ motif) from an Xray structure while this is not present in their map (since the sample studied is a chimera).

Yes, this point has been clarified in our revised Figure 5, which provides a visual guide to the modelling of the lever arm. Distinct colors are used to label the portion of the model that is not present in our map. Also, we have expanded the Materials and methods section to provide more details on how the extended lever arm models were constructed (subsection “Building atomic models with MDFF”, last paragraph).

How good is the density to properly orient the converter and to describe its conformation? This should be presented in a zoom of this region (movie), directly comparing the position of the density in rigor versus strong ADP.

In our revision we show the atomistic models of the converter and lever arm fit into their respective densities in Figure 5, Figure 5—figure supplement 1, and Video 3. We believe the density quality is sufficient to properly orient the converter and describe its conformation in both states. We also believe the density is sufficient to distinguish between kinked and straight lever arm models, with clear density for the kinked lever arm in rigor vs. straight in ADP. We provide a zoom of this region (Figure 5—figure supplement 1) and a movie (Video 3) to directly compare the density between ADP and rigor, as the reviewers requested.

The elongation of a particular orientation via addition of the rest of the lever arm magnifies the changes which seem overall rather small in the region of the converter and inser-2/Cam and which also suffer from lack of continuous density.

The density for the insert-2/CaM region is less detailed than the actomyosin interface, but sufficient to orient this portion of the lever arm at sub-nanometer resolution. As shown in Figure 4 and Figure 4—figure supplement 1, we observe clear density for the insert-2 helix and CaM for both the ADP and rigor states.

There is no direct data supporting the orientation of the lever arm after the Cam bound to insert-2 for which the density is rather poor – thus the angle measurements provided seem speculative, or should be carefully presented.

As noted above, we have used coloring in our revised figure to highlight the portion of the model that is not present in our map, and we have been careful to describe the proposed structures as grafted models throughout the third paragraph of the subsection “MD rearrangements facilitating ADP release generate are accompanied by a lever arm bend which could be regulated by force”.

Note also that the reference to Lister et al. 2004 to validate the change between ADP and Rigor is not appropriate since the molecule studied under optical trap is the full myosin VI in which the lever arm is much longer than what is presented in Figure 4 – thus the 10 A coincidence between the model shown in Figure 4 and the optical trap data (Lister et al) is not appropriate and should be removed.

We thank the reviewer for pointing this out; a direct comparison is indeed not appropriate. However, we believe a large fraction of our readership will be single-molecule biophysicists who will be interested in knowing how our structural results compare with previous single-molecule measurements. We have re-written this section to make explicit the differences in the constructs employed in the studies (subsection “MD rearrangements facilitating ADP release generate are accompanied by a lever arm bend which could be regulated by force”, third paragraph).

3) The previously reported ADP release related lever arm swing in myosin V, which doesn't contain the insert 2 region, calls for greater caution in the interpretation of present results.

We have added a paragraph (subsection “MD rearrangements facilitating ADP release generate are accompanied by a lever arm bend which could be regulated by force”, first paragraph) to clarify the proposed mechanism and point out that myosin V is able to achieve a small lever arm reorientation upon ADP release without the benefit of this insert-2-dependent effect, as the reviewer notes.

4) It is also puzzling that a lot of previous publications on the role of specific sites on actin in the binding of myosin (+/- nucleotides), as examined by mutational and structural approaches (from Rubenstein, Thomas, Reisler, and other groups), including the publication on hydroxyl radical mapping (footprinting) of myosin II sites on actin (Otzug-Durer et al.) are not considered here. Because the authors don't expect significant variations in actin binding sites for different myosins, this weakness should be corrected in the revised manuscript.

We thank the reviewer for the specific suggestions of prior literature mapping the actomyosin interface for comparison to our structural data. We have surveyed mutational analyses of myosin-binding residues (primarily in yeast actin) from the labs noted by the reviewer, and we find most of this work has focused on acidic residues in the N-terminus of actin, e.g.:

Stark…Rubinstein, Lord. JBC. 2011. (PMID: 21757693)

McCane…Rubinstein. JBC. 2006. (PMID: 16882670)

Hansen…Rubinstein, Reisler. Biochemistry. 2000. (PMID: 10677229)

Solomon…Rubinstein. JBC. 1988. (PMID: 3198644)

Cook…Rubinstein. JBC. 1992. (PMID: 1349604)

As we do not visualize an ordered conformation of the N-terminus of actin in our reconstructions, we do not believe it is appropriate to include this category of references.

Additional residues reported to affect actomyosin in gliding filament assays such as yeast actin residues E99A/E100A (corresponding to residues E99/E100 in our model), reported in Miller…Rubinstein, Reisler, Biochemistry, 1996 (PMID: 8987990), are not in a position to interact with the myosin motor domain in either of our strong-bound actomyosin VI reconstructions (nor are they in the Raunser structure of actomyosin IIc). We think these residues are likely relevant for weakly-bound, intermediate states, which have not yet been visualized, consistent with the conclusions of this and related studies from the Thomas lab (Prochniewicz and Thomas, Biochemistry, 2001 (PMID: 11705383). Actin residues E24/E25 (D24/D25 in our model), also reported in Miller…Rubinstein, Reisler, Biochemistry, 1996, are in a position to feasibly interact with myosin loop 2 (which we do not visualize in our density maps due to disorder), and we now include a reference to this paper when we discuss these residues (subsection “Interactions at the myosin-actin interface are distinct between different classes of myosins”, fifth paragraph).

Several mutagenesis studies have included the strong-binding actin mutant I341A (corresponding to residue I341 in our model), originally reported in Miller…Reisler, Biochemistry, 1996. (PMID: 8619986). We indeed observe this residue contributing to the hydrophobic interface with the myosin HCM, and now include this reference in the Results section discussing this interface..

Additionally, we have now indicated explicit areas of consonance between our results and the Oztug-Durer foot-printing study (subsection “Interactions at the myosin-actin interface are distinct between different classes of myosins”).

We do note that there are several EM structures of actomyosin complexes in the literature, which we have referenced. We now emphasize that these structures do broadly report a consistent interaction surface with actin, as expected (subsection “Interactions at the myosin-actin interface are distinct between different classes of myosins”). However, for a detailed comparison at the residue level, we believe it is appropriate to focus on the other highest-resolution actomyosin structure (myosin IIC), which contains high-confidence sidechain-level information.

We believe a comprehensive comparison of structural studies of diverse actomyosin interfaces would most appropriately be the subject of a future review, particularly as more and more high-resolution reconstructions of divergent myosins bound to actin are likely to emerge in the next several years.

5) What differs between Strong ADP and Rigor are the main new data presented in this paper so it should be the main focus of the result presentation, avoiding speculative statements and clearly describing these differences and their link to the presence or absence of ADP in the molecule.The authors should rewrite the comparison between Strong ADP and Rigor and be conservative about what is clearly demonstrated by their new EM data and what comes from a speculation pushing its interpretation. When it comes to speculation, this should be described in Discussion rather than the Results and more caution should be clearly stated.

We thank the reviewers for the suggestion to focus on this comparison. To clarify the experimental support for our conclusions, we have supplemented all comparisons of regions between ADP and rigor with the cryo-EM density maps in addition to the MDFF models. Additionally, we have revised the language in the manuscript to clarify what is speculative vs. clearly shown by the density maps (Results section).

To validate the model of the Strong ADP map interpretation, since it is of lower resolution, it is critical to compare different models one would obtain upon starting from different myosin VI models (post-rigor or rigor-like, in addition to the start as PiR already presented). Whether the final models after fitting would all converge in critical regions that differ between the Strong ADP and Rigor states or whether they would end up showing different interpretations need to be described. Regions that vary should indicate either a lack of restraint from the EM map, which could be linked to variability of the region, or just lack of resolution of the current data.It would be important to describe whether there is sufficient information to place the Calpha (and/or side chain) unambiguously whenever the authors use the differences to suggest consequences in motor function (strong ADP/Rigor transition and actin structure changes during the powerstroke).

We agree with this cautionary note, and we had in fact included such a validation comparison in our original manuscript. We now expand upon this analysis, which was originally reported in the text only, in Figure 3—figure supplement 2 and Figure 3—figure supplement 6. To validate the model of the ADP map, we generated an ADP model built from the rigor-like (2BKI) as an initial model for MDFF, as detailed in the manuscript. We find that both ADP models, built from PiR (4PFO) or rigor-like (2BKI), are more similar to each other than either starting model, which validates the fitting of our models and our MDFF procedure. Furthermore, this analysis also suggests a detailed model of ADP occupying the nucleotide-binding pocket is not necessary to capture the protein rearrangements in this region at the backbone level. In Figure 3—figure supplement 2 and Figure 3—figure supplement 6, we provide a local RMSD analysis of this comparison and find that all models converge in critical regions that differ between ADP and rigor. Additionally, we find that the R561-Y91 cation-pi interaction is also absent from the ADP model when utilizing 2BKI as the starting model.

The detailed descriptions at atomic level (actin interface, in particular for the Milligan cation-pi interaction that would differ in Rigor and Strong ADP states should only be included if the EM density gives more confidence that the data support such atomic descriptions.

As mentioned above in Essential revision 2, we have provided density map figures to support our model of the Milligan contact R561-Y91cation-pi interaction.

6) How can the authors guarantee that the Strong ADP data used correspond to mainly one state despite the lack of resolution in particular for the regions that are distal (relay / SH1 helix / lever arm)?

In general, two types of structural heterogeneity may contribute to lower resolution in distal regions: heterogeneity produced by the superposition of discrete states, and continuous heterogeneity, representing some level of conformational flexibility around an average structure. We favor the latter, for the following reasons. We observe a continuous gradient of resolution from the core of the actin filament out to the distal end of Insert 2 (Figure 1—figure supplement 1). Low-pass filtering the ADP map to 7.5 Angstroms allows unambiguous positioning of the lever arm in this density, which is consistent with structural fluctuations around this average position. Indeed, revealing such average positions by low-pass filtering a map is a hallmark of continuous heterogeneity, and is consistent with previous expectations of conformational heterogeneity in the ADP state based on simulation and biophysical studies. It therefore likely that an atomic-resolution “snapshot” of the converter and lever arm of myosin VI would be very difficult to achieve without substantial advances in image processing, particularly in the ADP state, even with a substantially larger dataset utilizing better instrumentation.

We have now included an explicit sentence regarding potential sources of the lower resolution of the ADP reconstruction in the text (subsection “A unique contact is established upon transition from the ADP state to the rigor state”).

What limits the resolution of the current studies?

We believe a major limiting factor for the resolution of the ADP state structure is the number of images collected and corresponding number of segments used for 3D reconstruction (discussed in the next section in more detail). Obtaining high-quality grids with completely decorated filaments in the ADP state was substantially more challenging than the nucleotide-free state, due to the lower affinity of the myosin for F-actin in this state. Since we were limited in the number of images we could collect per session due to using an F20 / Gatan 626 holder, we were ultimately unable to obtain as many high-quality micrographs for the ADP state. As shown in Table 1, approximately 1/2 as many movies and 2/3 as many segments were used for the ADP reconstruction as the rigor state reconstruction. We thus believe this reconstruction may be, to some extent, signal limited by the number of segments.

In general, we believe the major limitations on the resolution of our study were:

a) Intrinsic heterogeneity of the specimen (discussed in previous section)

b) Relatively limited dataset size imposed by the length of sessions we could conduct with an F20 microscope and Gatan 626 cryo holder, as well as the optical quality and stage stability afforded by this instrumentation.

In defense of assertion b, we provide FSC curves for an unpublished reconstruction (from a different filament specimen) we have recently obtained from a 300 kV Titan Krios microscope since moving to Rockefeller (Author response image 1). These data were collected in super resolution mode on a K2 detector with a slightly smaller pixel size (1.0 Angstroms vs. 1.27). Otherwise, identical software was used to obtain the final reconstruction: Frealign v9.11, working with dose-weighted segments generated with per-particle alignments from alignparts_lmbfgs.

We calculated reconstructions from ~900,000 ASUs, utilizing both frames 2-50, corresponding to a cumulative dose of ~80 electrons, as well as frames 2-10, corresponding to a cumulative dose of ~16 electrons/Å2, a “traditional” low dose (in both cases, the full video was used for alignment). Reconstructions with nearly identical reported resolution of 3.6-3.7 Å were obtained, with essentially identical density quality by inspection (not shown). Indeed, the full-dose reconstruction has more signal across all resolution shells, suggesting incorporating signal from more frames improves the map. This suggests that, at least to around 3.5 Å resolution, dose-weighting is quite effective, with minimal benefit to be gained by excluding frames (and perhaps a small price to pay). Furthermore, these unpublished results suggest that our image processing pipeline is not a limiting factor in the resolution of the reconstructions presented in this study.

**Author response image 1. respfig1:** FSC curves for an unpublished reconstruction obtained from a 300 kV Titan Krios. Reconstructions calculated utilizing both frames 2-60 (green), corresponding to a cumulative dose of ~80 electrons, as well as frames 2-10 (red), corresponding to a cumulative dose of ~16 electrons/Å^2^, resulted in nearly identical resolution assessments of 3.6-3.7 Å. The full-dose reconstruction has more signal across all resolution shells, suggesting incorporating signal from more frames improves the map.

More detail should be provided about motion correction and the selection of frames used to calculate the EM maps.

We have now provided additional detail in the experimental methods about motion correction for frame alignment. We use Unblur for initial full-frame alignments, followed by per-particle drift correction and dose-weighting with alignparts_lmbfgs of Rubinstein. We used all frames collected (24) to calculate the EM maps and did not exclude or omit any frames.

How can the authors ascertain that the lack of resolution for the Strong ADP is not linked to population of different conformers which should thus raise caution in the interpretation of the map? The density of the converter and the insert2/Cam allowing the building of the model is poorly described as well as the limitations that result from it. The authors should be careful in avoiding over-interpretation based on careful description of the quality and validation of the model in different parts of the molecule. In this sense, they should also describe more carefully whether there are changes at the actin/myosin interface that would be associated also with this transition. If they don't have the resolution to say so, (since the HCM loop and loop2 density is missing from their map), they should state that higher resolution of the reconstruction is likely required to access these important details of the actin/myosin interface.

This is a point well taken, and potential reasons for the lower resolution of the ADP reconstruction were detailed in the previous 2 sections. Despite lower resolution in the distal portions of the map, we are able to unambiguously place secondary structural elements (Insert2 / Cam) in this region of the 7.5 Å filtered maps, which is sufficient for an overall description of the conformational changes we elaborate in the text and the conclusions we draw from them.

We have revised the text to advance our descriptions of interactions at the actomyosin interface with appropriate caution, and explicitly state when higher-resolution maps would be required to definitively establish specific side-chain level interactions (subsection “Interactions at the myosin-actin interface are distinct between different classes of myosins”).